# Non-equilibrium dynamics of a nascent polypeptide during translation suppress its misfolding

Lisa M. Alexander[1], Daniel H. Goldman[1,7], Liang M. Wee[2,3,4] & Carlos Bustamante[1,2,3,4,5,6]

Protein folding can begin co-translationally. Due to the difference in timescale between folding and synthesis, co-translational folding is thought to occur at equilibrium for fast-folding domains. In this scenario, the folding kinetics of stalled ribosome-bound nascent chains should match the folding of nascent chains in real time. To test if this assumption is true, we compare the folding of a ribosome-bound, multi-domain calcium-binding protein stalled at different points in translation with the nascent chain as is it being synthesized in real-time, via optical tweezers. On stalled ribosomes, a misfolded state forms rapidly (1.5 s). However, during translation, this state is only attained after a long delay (63 s), indicating that, unexpectedly, the growing polypeptide is not equilibrated with its ensemble of accessible conformations. Slow equilibration on the ribosome can delay premature folding until adequate sequence is available and/or allow time for chaperone binding, thus promoting productive folding.

[1] Department of Chemistry, University of California Berkeley, Berkeley, CA 94720, USA. [2] California Institute for Quantitative Biosciences, University of California Berkeley, Berkeley, CA 94720, USA. [3] Howard Hughes Medical Institute University of California Berkeley, Berkeley, CA 94720, USA. [4] Molecular Biophysics and Integrated Bioimaging Division, Lawrence Berkeley National Laboratory, Berkeley, CA 94720, USA. [5] Department of Physics, University of California Berkeley, Berkeley, CA 94720, USA. [6] Department of Molecular and Cell Biology, University of California Berkeley, Berkeley, CA 94720, USA. [7] Present address: Dept. of Molecular Biology and Genetics, Johns Hopkins University School of Medicine, Baltimore, MD 21218, USA. Correspondence and requests for materials should be addressed to C.B. (email: carlosb@berkeley.edu)

Through folding, polypeptides adopt the specific three-dimensional structures required for their function. In the cell, some of these structures begin to form co-translationally[1–4]. However, the overall stability of the folded protein is reduced with respect to the folded state off the ribosome[5–7] and the rate of folding can be slower[8]. Such suppression of folding rates during vectorial synthesis by the ribosome is thought to increase the amount of active protein by promoting folding in a domain-wise manner and by preventing off-pathway misfolded states[9,10].

Studies have shown that smaller domains can fold inside the ribosome exit tunnel starting when the ribosome has decoded 25 amino acids downstream of the domain boundary[11,12], whereas larger nascent domains require synthesis of ~ 40 amino acids past the domain boundary to fold[5,13,14]. Overall, these results suggest that the nascent chain can fold as soon as there is enough space in the tunnel or at the ribosome surface such that the folded state is not sterically precluded[15]. Accordingly, most co-translational folding studies employ stalled ribosome-bound nascent chain complexes (RNCs), which also ensure a uniform population during analysis[6,8,12]. However, whether the stalling is permanent (e.g., generated by mRNA truncation) or temporary (e.g., induced by ribosome stalling sequences such as SecM), the implicit assumption has been that the observed folding dynamics faithfully reflect the dynamics that occur during synthesis, as the transitions are thought to be fast relative to the rate of translation and the folding process is presumed to occur at equilibrium[16,17]. Although this assumption may be justified in some cases[18] it presupposes that the rates of elongation and folding are the only relevant ones controlling the overall behavior of the nascent chain. In fact, other processes may occur contemporaneously and the chain may not begin to fold immediately upon synthesis.

To test this hypothesis, we investigate the folding of a multi-domain protein, calerythrin, on an actively translating ribosome and compare it to the protein's folding on the surface of stalled RNCs and its folding off the ribosome. We show on stalled complexes that the ribosome can decrease the kinetics of folding and increase the unfolding rate of an off-pathway misfolded state. We also demonstrate that the ribosome reduces the stability of an N-terminal intermediate, precluding its formation during active translation, despite the fact that the N terminus is the earliest independently folding unit to emerge from the ribosome. Strikingly, we find that the folding pathway during active elongation is distinct from that derived from our stalled RNC data. Specifically, calerythrin exhibits non-equilibrium folding on actively translating ribosomes, whereby the onset of co-translational folding is delayed, suppressing the misfolded state. Together, our results indicate that a co-translational folding pathway is not necessarily determined by the rates of elongation and equilibrium folding alone, and that non-equilibrium contributions can enhance the probability of proper native folding.

## Results

**Full-length calerythrin folds through the C domain.** To investigate the folding of a multi-domain protein in solution, on stalled RNCs, and during synthesis, we used an optical tweezers (OT) assay that enables us to selectively denature the polypeptide (using force) while leaving the ribosome intact. With this assay, we can directly detect folding intermediates and measure the rates of folding and unfolding. We selected as a model protein calerythrin, a two-domain calcium-binding protein from the bacterium *Saccharopolyspora erythraea*[19,20]. Calerythrin is a member of the EF-hand family of proteins. It contains four EF hands, each of which consists of a helix-loop-helix motif with a calcium-binding site in the loop. One pair of hands (EF1 and EF2) forms the N

domain and another pair (EF3 and EF4) the C domain (Fig. 1a). Nuclear magnetic resonance and circular dichroism studies suggest that the C domain folds more quickly than the N domain upon addition of calcium[21]. EF2 is noncanonical and does not bind calcium[22]. Other members of the EF-hand family have been previously studied using OT and these studies have yielded detailed measurements of folding networks off the ribosome[23–25]. In order to provide a benchmark to interpret our real-time elongation results, we first characterized the folding of calerythrin off the ribosome and the equilibrium folding kinetics of stalled RNCs.

We tethered the isolated full-length (FL) protein by attaching the N and C termini to DNA handles (Fig. 1a, b) and collected data via force–extension measurements, where the traps move apart at a fixed rate. In this measurement, applying load (increasing force) leads to unfolding of calerythrin (detected as increasing extension). In a typical force–extension curve, at 13 ± 1.3 pN (s.d.), FL calerythrin unfolds in a single rip, indicating cooperative unfolding (Fig. 1c). The unfolding transition is 65 ± 2 nm, or 177 ± 5 amino acids, as expected for the full unfolding of calerythrin, which is 177 amino acids in length. Although both domains unfold in one transition, the domains refold sequentially in two transitions (Fig. 1c). To investigate the refolding in more detail, we collected data in passive mode (where the traps remain stationary, as contrasted with force–extension measurements where the traps move apart at a fixed rate). In this mode, the force suddenly increases if folding occurs. The passive data show hopping between unfolding and an intermediate state before FL folding (Fig. 1d). FL folding is long-lived at this force and must be unfolded at high force to observe further transitions. We used a Hidden Markov Model (HMM)[24,26] to extract the lifetimes of the unfolded and intermediate states as a function of force (Fig. 1e). Because the HMM with only one intermediate converges to single exponential distributions (Supplementary Fig. 1a) and this state's kinetics follow Bell's model as expected, we conclude that there is only one intermediate ($FL_{int}$), and the protein refolds every time through this same state before reaching the fully folded state (Fig. 1d, e).

Next, we measured the folding of each domain individually by preparing truncated proteins of either the N domain (amino acids 1–90) or the C domain (91–177) with the same tags on the termini. Folding of the isolated N domain involves more amino acids, is slower, and occurs at lower forces than $FL_{int}$ (two-tailed Welch's *t* test, $p < 0.05$) (Fig. 1c–e, Supplementary Table 1). In contrast, folding of the isolated C domain is indistinguishable from $FL_{int}$ in number of amino acids and the folding and unfolding kinetics (Fig. 1c–e, Supplementary Table 1), suggesting that $FL_{int}$ corresponds to the folding of the C domain.

To confirm this assignment, we individually mutated the conserved 12th position glutamate/aspartate to glutamine in the EF-hand loops of hands 1, 3, or 4, thus impairing calcium binding and destabilizing one EF-hand fold at a time[27,28] (EF hand 2 does not bind calcium in the natural sequence[29]). The glutamine mutation in EF1 (belonging to the N terminus domain) led to a significant change in the unfolding transition of the FL protein, which is no longer cooperative and the domains unfold in two sequential steps. However, this mutation does not affect the folding or unfolding kinetics of the first intermediate state, which is indistinguishable from the wild-type $FL_{int}$ (Supplementary Fig. 2a). In contrast, mutation of this residue in EF3 or EF4 destabilizes the intermediate so that it only folds at lower forces and unfolds more rapidly (Supplementary Fig. 2b, c). These mutant studies confirm that the $FL_{int}$ intermediate corresponds to the folded C domain. Accordingly, the folding pathway of the isolated FL molecule proceeds from the unfolded state through the C domain to the native structure. Folding through the C

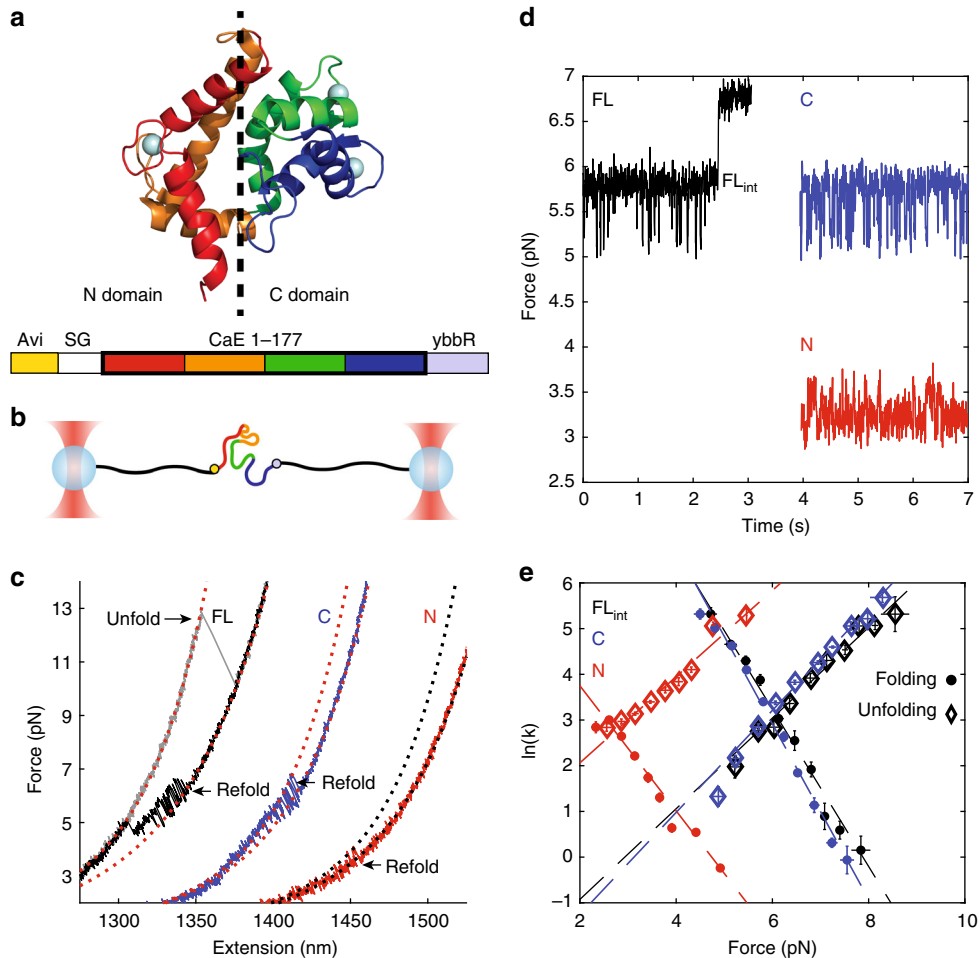

**Fig. 1** Full-length calerythrin folds through the C domain. **a** The NMR structure of calerythrin (PDB: 1NYA[20]). EF 1, 2, 3, and 4 are red, orange, green, and blue, respectively. The protein sequence contains an Avi tag at the N terminus and ybbR tag at the C-terminus. **b** The geometry in OT is shown, with 2 kb of DNA on each side of the protein. **c** Force–extension curves of the FL, N, and C domain samples, offset in extension. The WLC model of DNA+protein is shown as dotted red lines, except for N domain, which is dotted black. The protein WLCs correspond to lengths of 177, 75, and 90 amino acids for FL, C, and N. The FL protein unfolds cooperatively (gray) and refolds in two steps (black). The C (blue) and N (red) domains fold reversibly in one step; for clarity only the refolding curves are shown. **d** Passive of the FL, C, and N constructs (offset in time). In all cases, the lowest observed force is the unfolded state, and then transitions are observed to a folded state. For FL (black), the first transition is $FL_{int}$ and the highest transition is the full-folded structure. The N and C fold to their one-domain structures. **e** The kinetics of folding (circles) and unfolding (diamonds) for $FL_{int}$ match the kinetics of the C domain but are markedly different than the N domain. Error bars are standard error (SE). $FL_{int}$ $n = 11$ molecules, 116 rate measurements; C domain $n = 9$ molecules, 107 rate measurements; N domain $n = 13$ molecules, 100 rate measurements. See Supplementary Table 1 for the fits of the Bell model (dashed lines). See Supplementary Fig. 2 for further confirmation of $FL_{int}$ identity. Source data are provided as a Source Data file

domain exclusively, rather than through either domain as in calmodulin, has been reported for another EF-hand protein NSC1[25], which like calerythrin also has a noncanonical EF hand in the N domain.

**N domain folding is suppressed on the ribosome.** Having established the off-ribosome folding pathway of calerythrin, we next sought to characterize how the surface of the ribosome modifies the folding of the protein in stalled complexes, wherein the process occurs at equilibrium. We prepared RNCs stalled at codons 135–177 in 10–11 codon increments (Fig. 2a) and tethered them via the N terminus of the nascent chain to one DNA handle and a ybbR tag on protein L17 of the ribosome to the other DNA handle (Fig. 2b)[8]. At codon 135, the N domain (1–90, corresponding to EF1 and EF2) is completely outside of the exit tunnel[30]. However, we do not observe folding for this complex (RNC135), nor for RNC146 or RNC157 (Fig. 2c, Supplementary Fig. 3). The isolated N domain off the ribosome folds at a rate of

$k_0 = 660 \pm 370\ \mathrm{s}^{-1}$, so we infer that the ribosome destabilizes the N domain by reducing its folding rate or increasing its unfolding rate relative to the isolated N domain such that it is not sampled. A decrease in folding rate of up to two orders of magnitude has been reported for other proteins such as T4 lysozyme on the ribosome, although in that case folding was reduced but not eliminated[8]. The N domain intermediate observed here has lower stability than many full proteins or domains, and the destabilization may be sufficient to prevent folding entirely. Thus, even though the N domain is readily sampled for the isolated protein off the ribosome, and the vectorial N-to-C nature of protein synthesis precludes early formation of the C domain, folding on the ribosome does not proceed through an N-terminal intermediate.

**Stalled RNCs misfold.** We next extended the polypeptide further to generate RNC167 and RNC177 (RNC177 is FL but with part of the C domain in the tunnel). Surprisingly, although we still failed

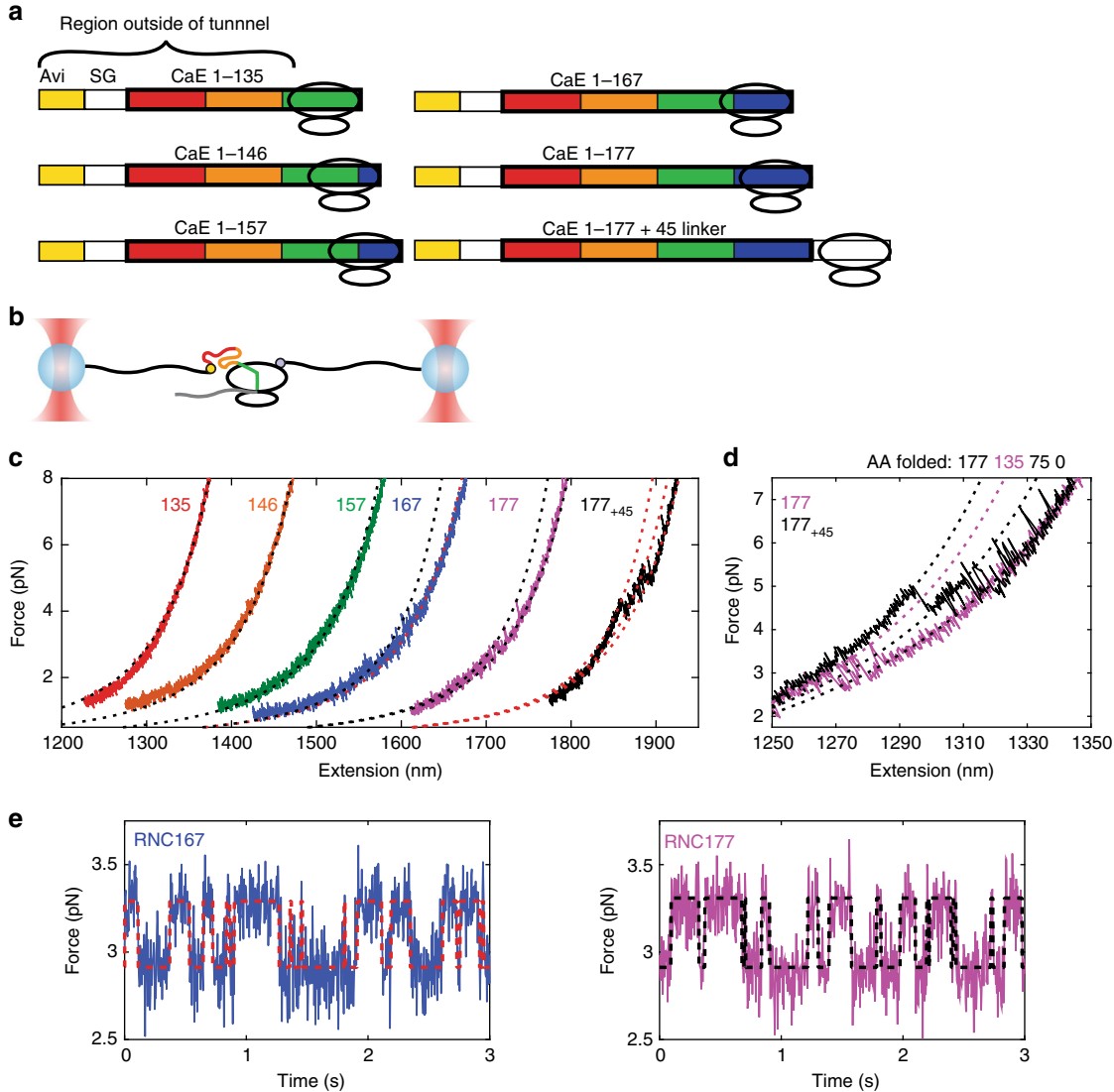

**Fig. 2** Stalled RNCs misfold. **a** The construct design for stalling. Using truncated mRNAs that lack a stop codon, we are able to generate homogeneously stalled ribosomes. The last codon added (in the P site) corresponds to the name of the RNC; less sequence than this will be outside the tunnel. **b** The OT geometry uses an N-terminal Avi tag and a tag on the ribosome itself. **c** Refolding force–extension curves of our RNCs do not show any folding until codon 167 or 177. WLC models are shown as dotted black or red lines. The WLC model assumed 40 amino acids in the ribosomal exit tunnel. For RNC167 and RNC177, folded WLCs show a transition of 135 amino acids. For RNC177$_{+45}$, folded WLCs correspond to a transition of 75 and 177 amino acids. See Supplementary Fig. 3 for further testing. Representative force–extension curves from $n = 3, 10, 9, 5, 10, 5$ molecules, respectively. **d** An example of the folding of RNC177 and RNC177$_{+45}$. RNC177 has a misfolded state larger than the C domain (75 amino acids) but smaller than the FL protein (177 amino acids). RNC177$_{+45}$ folds to the FL state. The dotted lines are WLC models to 177, 135, 75, and 0 amino acids folded (black, magenta, black, black, respectively). **e** Passive mode data of RNC167 and RNC177 shows folding transitions to state M. HMM fit shown as dashed lines (red for RNC167 and black for RNC177). Source data are provided as a Source Data file

to observe folding of the N domain, we observed a misfolded state, M, of $130 \pm 15$ amino acids, significantly larger than either the N or C domains alone, for both RNC167 and RNC177 (Fig. 2c–e, also compare to Fig. 1d and Supplementary Table 1). Only by extending the entire protein sequence outside of the exit tunnel with a 45 amino acid artificial linker (RNC177$_{+45}$) is native-like folding restored (Fig. 2c, d). At this stage, the misfolded state is not observed because native folding through the C domain occurs at higher forces and is more stable, outcompeting the misfolded state. These observations explain why the misfolded state is not seen in isolated FL protein experiments.

**The misfolded state has altered kinetics on the ribosome.** Although the misfolded state observed with RNC167 and

RNC177 is not observed for the fully synthesized and released protein, it is accessible to a truncated version of the protein lacking the 4th EF hand (EF123, amino acids 1–134), owing to the lack of sequence required for C domain folding. The isolated EF123 can fold to either state M or the N domain in solution (Fig. 3a, Supplementary Fig. 4). Moreover, the misfolded and folded N domain states of EF123 do not interchange without first reverting to the fully unfolded state, as shown from our HMM analysis and analysis of the point-spread function[31] (Supplementary Fig. 4). This observation suggests that the misfolded state is off-pathway for the nascent chain, involving non-native interdomain contacts that preclude the correct folding of the individual domains. Misfolding of EF-hand proteins has been reported previously, either through mispairing EF hands or via

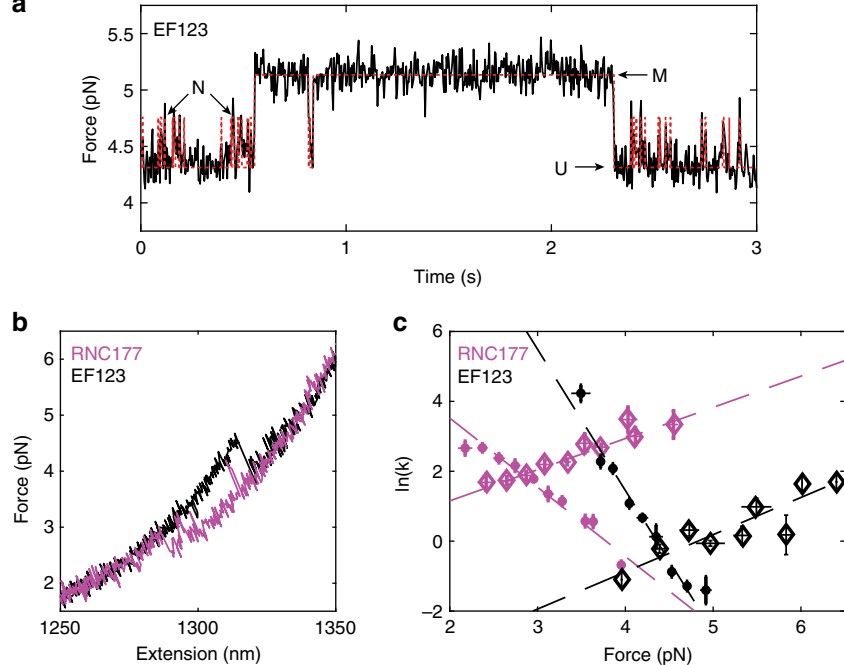

**Fig. 3** The misfolded state has altered kinetics on the ribosome. **a** Passive data for EF123 shows a short-lived state (N domain) in between the two more distinct states (U, unfolded, at lowest force and M, misfolded, at the highest force). The red dashed line is the HMM fit. **b** Overlaying the refolding curves from RNC177 (magenta) and EF123 (black) shows they are the same size transition, although EF123 has short-lived N transitions prior to full misfolding. **c** The folding kinetics of RNC177 are slower and the unfolding kinetics are faster relative to EF123. Diamonds: unfolding rates, circles: folding rates. Error bars are standard error (SE). See Supplementary Table 2 for the fits from the Bell plot shown. RNC177: $n = 10$ molecules, 111 rate measurements. EF123 (unfolding): $n = 14$ molecules, 97 rate measurements. EF123 (folding): $n = 13$ molecules, 88 rate measurements. Source data are provided as a Source Data file

off-pathway compaction of partially structured intermediates[24,28]. As for calerythrin, since the misfolded state forms directly from the unfolded state and involves the entire available sequence of the truncated protein, state M could involve mispaired EF hands, similar to a previously reported misfolded state of calmodulin[32]. EF2 is also a noncanonical EF hand that does not bind calcium[22], potentially allowing EF3 folding to preempt EF2 folding. Thus, although the FL protein off the ribosome usually avoids the misfolded state owing to the faster folding of the C domain, M is accessible on the surface of the ribosome when the sequence needed to form the C domain is still partly sequestered in the ribosomal exit tunnel and the ribosome has destabilized the N domain (as shown with RNC135).

The misfolding occurs at much lower forces on the ribosome than on the isolated polypeptide, as observed in the overlaid force–extension relaxation curves of RNC177 and EF123 (Fig. 3b). Significantly, the ribosome decelerates the folding kinetics of the misfolded state at codons 167 and 177 by $10^4$-fold and accelerates the unfolding rate by 90-fold at zero force compared with the free truncation EF123 (Fig. 3b, c, Supplementary Table 2). These trends reduce the probability that the RNC attains the misfolded state and increase the probability that the molecule escapes it to recover productive folding if it does form. Notice, however, that the time until misfolding on the ribosome at zero force, whereas longer than for the free EF123 construct, is still quite short ($6 \pm 3 \times 10^{-4}$ s) and the time it remains trapped in this state ($1.9 \pm 0.5$ s) is still relatively long. Assuming that our equilibrium rates on stalled RNCs and reported cellular rates of elongation ($4$–$6\,\text{s}^{-1}$ at room temperature[33]) are the only determinants of co-translational folding, the misfolded state would be expected to form readily during translation in spite of the relative destabilization by the ribosome. In real-time elongation conditions, assuming the rates we have measured do not change

substantially with chain length (as is the case between codons 167 and 177), we would expect on average one misfolding event per 1.5 sec at 4.0 pN.

**Translating RNCs adopt non-equilibrium trapped states.** Next, we asked: are the kinetics to-and-from the misfolded state—as determined at equilibrium on the stalled RNCs—predictive of the order of folding events during active translation? To answer this question, we needed to follow the growth of calerythrin in real-time while simultaneously monitoring its co-translational folding. Accordingly, we generated stalled RNCs by amino-acid omission (Supplementary Fig. 5), and restarted peptide elongation by introducing an in vitro translation mix (Fig. 4a, Supplementary Table 3)[34]. The mRNA was engineered to contain 225 codons to translate in the tweezers: a three-valine stall site, all 177 amino acids of native calerythrin, and the same C-terminal linker used for generating the stalled RNC177+45 (see Fig. 2a). As these experiments are conducted in passive mode, active elongation is observed as a drop in force concomitant with the polypeptide length increase (Fig. 4b). Folding transitions appear as a sudden increase of the tension across the tether. We conducted translation at 4.5–3.5 pN, so that it is in the similar range of forces as probed using stalled RNCs.

We converted the change in force to amino-acid position using a worm-like chain (WLC) model for the polypeptide[35] and an extensible WLC for the DNA handles[36] (Fig. 4c). After this conversion, translation is observed as an increase in the number of tethered amino acids, and folding as a sudden drop when those amino acids are sequestered. By fitting the data with a general isotonic regression[37,38], we find an average elongation rate of $1.0 \pm 0.3$ amino acids/s, consistent with other in vitro translation experiments using the commercial PURExpress system[39,40]. We

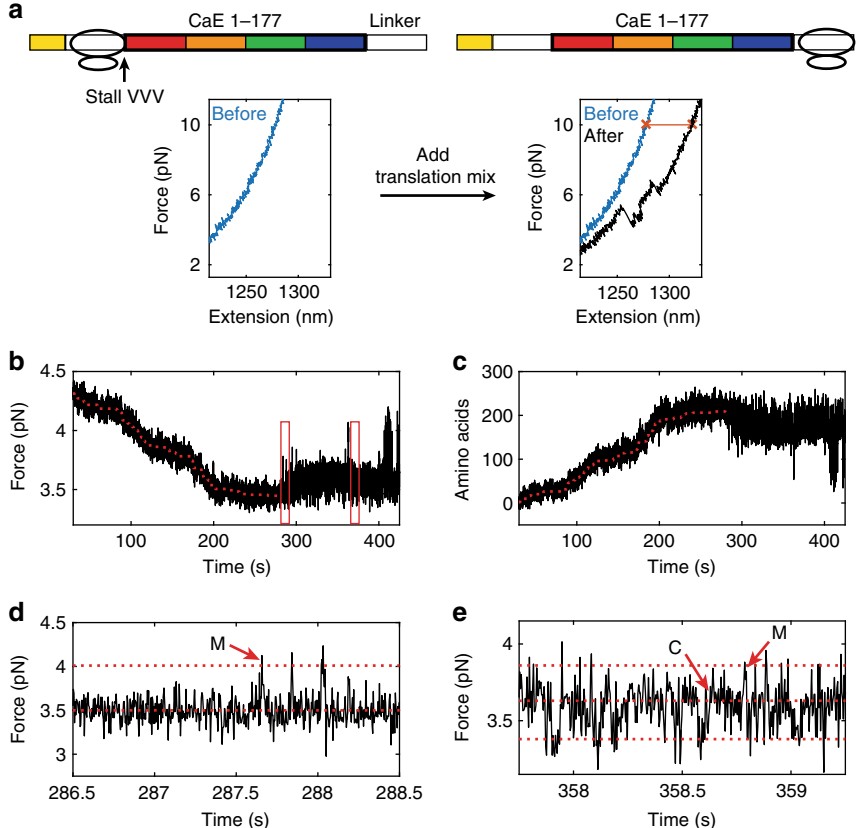

**Fig. 4** Set-up for real-time elongation OT experiments. **a** The sequence for stalling and restart. RNCs were stalled at a three-valine repeat (see Supplementary Fig. 5) and restarted with the addition of translation mix in the OT (Supplementary Table 3). The force–extension curves show the lengthening of the tether after translating (red line denotes difference between blue and black force–extension curves) as well as folding of the full-length protein, which confirms the ribosome reached the end of the mRNA. **b** An example of ribosome translating. As each amino acid is added, the force decreases as the tether gets longer. The fit is shown as a red dotted line. **c** The trajectory is converted to amino acids using a worm-like-chain model. **d** A zoom of the panel in **b** that shows the folding region. The red arrows show the misfolded state and the dotted line the expected size for state M. **e** A zoom of the panel in **b** that shows the folding region later in time. At this point, both C and M are accessible (dotted lines show expected sizes, arrows mark some the transitions). Representative molecule from $n = 12$. Source data are provided as a Source Data file

point out that our in vitro system does not perfectly recreate the cellular environment— for example, it does not include chaperones and does not mimic the extent of molecular crowding in the cell.

At forces below 5 pN, we observe transitions to the folded C domain or to the misfolded state (Fig. 4d, red arrows mark the M state or C domain). Every molecule exhibits misfolding or C domain folding, but N domain folding is not observed, as expected from the stalled RNC data (Fig. 2c). Most surprisingly, however, although the misfolded state is detected during real-time translation, it does not form readily when the ribosome has reached codon 167, as would be expected from our experiments with equilibrated stalled complexes. Instead, we observe a delay until the onset of folding, after which the chain (mis)folds and unfolds multiple times (hopping) as previously observed with the stalled RNCs (Fig. 5a–c; the delay time $\tau$ is indicated, at the end of which the hopping is observed).

We measured the initial delay as the time elapsed between the moment when the ribosome reaches codon 167 to the time when we first observed a folding transition (Supplementary Fig. 6), either to the misfolded state (75% of the molecules) or to the folded C domain (25%) (Fig. 5b, c). In 8% of molecules, the delay is essentially indistinguishable from the lifetime of the unfolded state in stalled complexes ($\tau_{unfold}$), demonstrating that this outcome is observable in our assay, though not very common. The distribution of measured $\tau$ values is exponential

(Supplementary Fig. 7), with the characteristic delay until the onset of folding $\tau_{delay} = 63 \pm 12$ s from the fit to the cumulative distribution (Fig. 5d). This exponential distribution supports the idea that, as it is being synthesized, the polypeptide is kept out of equilibrium in the unfolded state until it undergoes a stochastic equilibration step. The characteristic delay is 40 times longer than the measured lifetime of the unfolded state at equilibrium in RNC167 and RNC177 ($\tau_{unfold} = 1.5 \pm 0.3$ s at 4.0 pN, the average force at which the elongating ribosomes reach codon 167). This delay is not owing to interactions between components of the translation mix and the nascent chain, nor to linker effects, as experiments with stalled RNCs in translation mix or added linkers did not display a delay (Supplementary Fig. 8).

This behavior indicates that prior to the onset of folding, the newly synthesized polypeptide resides in a trapped unfolded state that is not folding-competent. Although there is a proline in the sequence of the C domain, this trapped state is unlikely to be owing to isomerization as the proline in the NMR structure is in the trans configuration[20], the orientation in which amino acids are added at the peptidyl transfer center. Instead, this state may involve a set of local interactions with the ribosome surface or exit tunnel that were established during its growth in real time or, alternatively may form only in the context of actively elongating ribosome surface dynamics. For example, in real time, there will be local dynamics of ribosomal components as factors bind/ unbind and ribosomal elements alter conformation. It is also

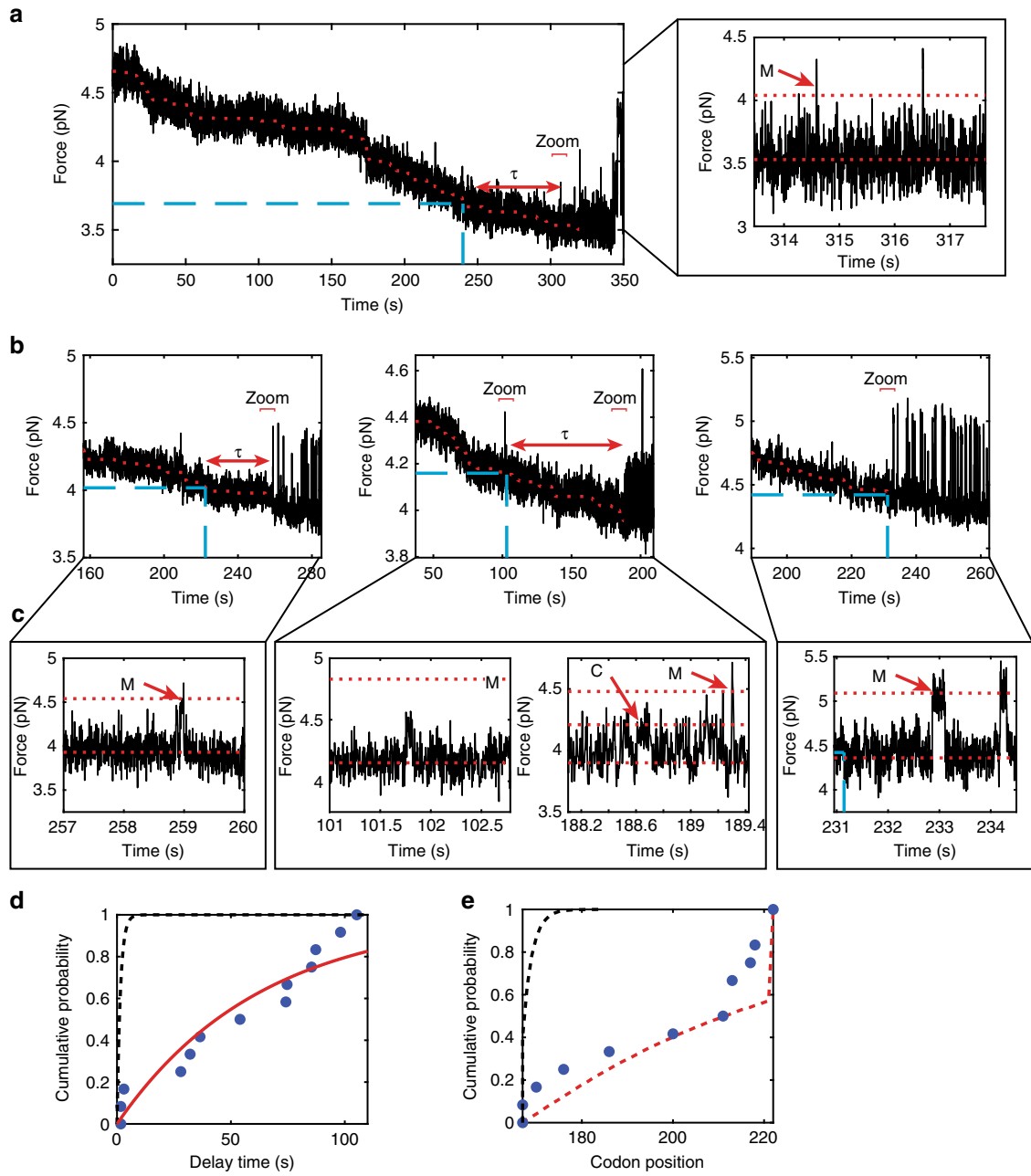

**Fig. 5** Translating RNCs have a non-equilibrium delay prior to folding. **a** An example trajectory showing full translation of the sequence in force versus time. The blue dashed line corresponds to codon 167, when folding would begin in equilibrium conditions. $\tau$ indicates the observed delay. The inset shows a zoom to the folding region shown at higher bandwidth (267 Hz) to see the folding transitions. **b** Example trajectories zoomed to show just the region after passing codon 167 (blue dashed line) to the onset of folding, either the misfolded state (1st example), the C domain (2nd example), or the misfolded state with an equilibrium-like delay (3rd example). **c** Zoom in regions of the data in **b** at higher bandwidth (267 Hz) show the folding transitions. For the spike in **b** in the middle panel at 101.5 sec, the zoom shows that this transition is a rare event that does not correspond to N, C, or M folding as it is too small. This event does not reoccur, and was not scored as the onset of equilibrium-like folding. **d** The observed delay time empirical cumulative distribution function (blue dots). An exponential fit is shown as a red line, whereas the expected distribution for time until folding based on simulations using stalled RNC data is shown as a black dotted line. **e** Using our fit to codon position, we can also record the codon where folding first occurred (blue dots). This is compared with a simulation of equilibrium conditions (black dotted line) and a simulation assuming an additional delay step of 63 sec (red dotted line). Source data are provided as a Source Data file

possible that the chain may form local structures when it emerges from the tunnel that are different from a random coil unfolded state, that may be folding incompetent. For example, small helical sections are known to fold in the tunnel[41,42] that may prevent the folding process. Once the chain manages to escape from this trapped state, however, it is seen to (mis)fold and unfold without

visiting again the trapped state, becoming equilibrated within its energy landscape. In support of this interpretation, once the misfolded state forms, ramping the force to unfold it and returning it to forces as low as 2 pN, we observe no subsequent delay; instead, hopping between unfolded and (mis)folded states resumes, as in stalled complexes (Supplementary Fig. 9).

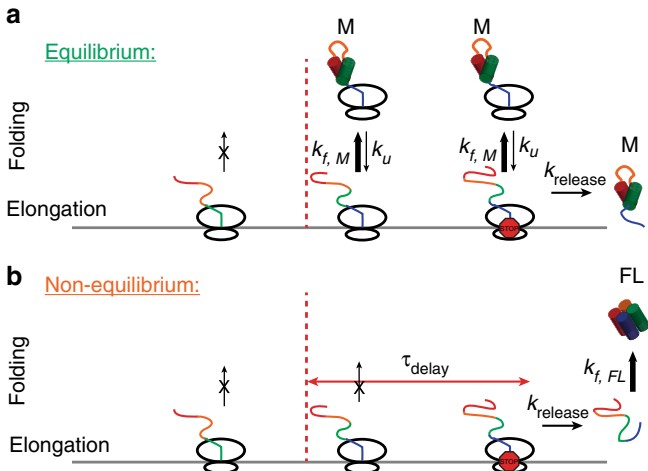

**Fig. 6** Schematic of co-translational folding in and out of equilibrium. **a** The equilibrium scenario shows the onset of folding after reaching the minimum length predicted from stalled complexes (dashed red line). After this point the nascent chain misfolds. At the stop codon, the misfolded protein would be released from the ribosome. **b** Out of equilibrium, as we observe, there is a delay after reaching the length required to fold seen in equilibrium. This delay is long enough that the unfolded protein is released upon termination, whereupon it would fold post-translationally

## Discussion

Since translation in our system is on a similar timescale (1.0 amino acids/s) as folding for calerythrin measured in the stalled RNCs under force (misfolds in 1.5 s at 4.0 pN, although this would be faster at zero force), we expected the kinetics of folding during real-time elongation experiments to display a quasi-equilibrium behavior that reflects that of stalled, equilibrated RNCs. Using our measured parameters, we performed simulations of translation and folding behavior under equilibrium and non-equilibrium assumptions. We would expect the first mis-folding transition to occur in 3.5 sec or less for 90% of ribosomes in equilibrium (Fig. 5d). In terms of codon position, we would expect >99% of RNCs to fold before codon 177 in real-time conditions (Fig. 5e). In this scenario, the RNC would not be able to avoid the misfolded state and the misfolded product would be released in solution (Fig. 6a). Yet, despite the relatively slow rate of translation in our experiments—rates that should even more strongly favor a quasi-equilibrium behavior because there is more time for folding at each subsequent amino acid added—we see that the nascent polypeptide folds only after a significant time delay. Given the known elongation rate of 12–18 amino acids/s in vivo at 37 degrees[33] and a measured $k_{cat}$ of 1.5 s$^{-1}$ for peptide release[43], the measured delay of 63 s is long enough to avert misfolding for the remaining duration of elongation until the release of the nascent polypeptide. After release, the nascent chain would fold through the previously described off-the-ribosome pathway (Fig. 6b). Such a long delay could also allow time for chaperones to bind to the unfolded polypeptide and prevent the formation of the misfolded state on the ribosome.

Kinetic modeling suggests that not all proteins are equilibrated on a translational timescale, and the percent of proteins that are predicted to fold co-translationally can vary from 37% to 59% of the proteome when altering translation speed[16]. However, experimental evidence that folding is not necessarily limited by the translation rate and folding takes place out of equilibrium has been indirect, for example, through decreases in protein activity after increasing the translation rate via synonymous codon mutations or changing tRNA levels[1,3,4,44]. Other in vitro experiments that incorporate active translation employ ensemble

assays where ribosomes are not synchronized[13], which could make deviations from equilibrium behavior difficult to detect, depending on the magnitude of the delay and the folding rates in question. Using a real-time single-molecule assay we detect a clear signature of non-equilibrium dynamics that may play an important role in the fate of the newly synthesized polypeptide. The polypeptide is not equilibrated with the ensemble of accessible conformations measured in stalled complexes; rather, it persists in the unfolded state until some equilibration step occurs. Our results show that comparing equilibrium-derived folding rates from stalled RNCs with translation rates is not always sufficient to predict co-translational folding of a polypeptide. Indeed, in this case we find that the peptide product must be observed in real-time as it is being generated, as other processes may also be at play that are obscured in stalled complex measurements. Stalled complexes require, for most experimental techniques, at minimum tens of minutes to prepare, which allows time for equilibration processes to occur. If these processes could be induced in stalled complexes, we would expect to be able to predict the co-translational behavior during continuous synthesis; however, as the exact nature of this process is unknown and was not previously predicted, the measurement must be conducted in real-time. The real-time folding assay developed in this study will also be useful in other situations where measurements of stalled RNCs may not capture the full dynamics of folding, such as length-dependent protein conformational changes, dynamic interactions with chaperones, or folding near domain boundaries.

## Methods

**Constructs and purification of proteins**. The calerythrin gene was purchased (UniProtKB—P06495, IDT gene block), amplified, digested (NotI, NheI from NEB), and inserted into a plasmid with flanking Avi and ybbR tags, with a serine–glycine linker after the AviTag (TSGGGGSGGGGSGGGGSGGGGS). Truncated constructs for released proteins were prepared via PCR and digestion with the same set of enzymes followed by cloning into the same backbone vector. Site-directed mutagenesis for calcium mutants and insertion of the valine stall site was conducted via whole-plasmid PCR amplification with mismatches in the primers, followed by DpnI and PNK treatment and blunt-end ligation with T4 Ligase. Constructs were transformed into DH5α cells (Fisher Scientific, 18265017) and plasmid DNA was harvested via mini-prep. All constructs were verified by plasmid sequencing. The primers for all cloning steps are listed in Supplementary Table 4.

Wild-type calerythrin, calerythrin truncations, and calerythrin mutants were expressed in BL21 DE3 *Escherichia coli* cells (NEB, catalog number C2527H) with isopropyl β-D-1-thiogalactopyranoside induction (1 mM for 2 hrs at 37 °C), lysed, and biotinylated. Biotinylation of the lysate was conducted with 1 μM recombinant birA, 25 μM biotin, 5 mM ATP, and 5 mM Mg(OAc)$_2$ at room temperature for 40'. Excess biotin was removed by overnight dialysis in 1× PBS with 0.5 mM phenylmethylsulfonyl fluoride (PMSF). Biotinylated protein was purified on monomeric avidin resin using standard elution procedures (Pierce). Excess biotin was removed by overnight dialysis in 20 mM HEPES, 100 mM KCl, and 0.5 mM CaCl$_2$. The protein sample was concentrated via Amicon centrifugal filters (Millipore, MWCO 10 kD or 3.5 kD) and stored at −80.

Purification of the tRNA synthetases was based on reported procedures[45,46]. In brief, each tRNA synthetase was auto-induced in and extracted from BL21 strains by sonication. The enzymes were then captured on nickel sepharose columns (HisTrap™ HP, GE Heathcare) in binding buffer (50 mM HEPES-KOH, pH 7.5, 1 M NH$_4$Cl, 10 mM MgCl$_2$, 25 mM imidazole and 7 mM β-mercaptoethanol) and isolated from the column using a linear gradient (0–100%) of elution buffer (50 mM HEPES-KOH, pH 7.5, 100 mM KCl, 10 mM MgCl$_2$, 1 M imidazole and 7 mM β-mercaptoethanol). Fractions that contained the enzymes were pooled, dialyzed into storage buffer (50 mM HEPES, pH 7.5, 100 mM KCl, 10 mM MgCl$_2$ and 7 mM β-mercaptoethanol), concentrated and supplemented with glycerol to 30% (v/v) before storage at −80 °C. Aspartyl-tRNA synthetase and glutaminyl-tRNA synthetase were further purified by anion-exchange (HiTrap Q HP, GE Healthcare) prior to dialysis in storage buffer. The buffers for ion-exchange chromatography consisted of 50 mM HEPES-KOH, pH 7.5, 10 mM MgCl$_2$, and 7 mM β-mercaptoethanol with either 50 mM (binding buffer) or 1 M (elution buffer) KCl. The gene for nucleotide diphosphate kinase (NDK) was amplified from bacteria strain MG1655 (The Coli Genetic Stock Center #6300, Yale) and cloned into pET His6 TEV LIC cloning vector (Plasmid #29653, Addgene) to introduce an N-terminal His-tag for affinity purification. Purification used the same binding and elution buffer for tRNA synthetases except that the starting final imidazole concentration in the binding buffer was increased to 40 mM. The purification of NDK ended with size exclusion chromatography (HiPrep™ 16/60 Sephacryl® S-100 HR, GE Healthcare) using

50 mM HEPES-KOH, pH 7.5, 100 mM KCl, 10 mM MgCl₂, and 1 mM dithiothreitol. NDK containing fractions were pooled, concentrated and adjusted to 30% glycerol (v/v) for storage at −80 °C. Pyrophosphatase, myokinase, and creatine kinase were purchased from Sigma. EF-G, EF-Tu, and EF-Ts were expressed with autoinducing media. EF-G was purified via a His column gradient from 20 mM to 300 mM imidazole in 1× PBS with 3 mM 2-mercaptoethanol over 12 column volumes (CV). The EF-G fractions were pooled, diluted threefold, and then further purified on a Q column via gradient from 50 mM to 600 mM NaCl in 25 mM Tris pH 8 with 3 mM 2-mercaptoethanol over 20 CV. EF-G was dialyzed into 25 mM Tris, 60 mM NH4Cl, 6 mM 2-mercaptoethanol overnight, concentrated and stored at −80. EF-Ts was purified in a gradient from 10 mM to 1 M imidazole in 20 mM Tris, 300 mM NaCl, and 2 mM 2-mercaptoethanol over 12 CV. Fractions were pooled and dialyzed overnight into 50 mM HEPES, 100 mM KCl, and 10 mM 2-mercaptoethanol. The purified protein was concentrated and supplemented with buffered glycerol to 30% final concentration (buffer composition unchanged), then stored at −80. EF-Tu was purified in a gradient from 30 mM to 800 mM imidazole in 20 mM Tris, 300 mM NaCl, 0.5 mM MgCl2, 50 μM GDP, and 2 mM 2-mercaptoethanol over 12 CV. Fractions were pooled and dialyzed overnight into 50 mM HEPES, 100 mM KCl, 0.5 mM MgCl2, 50 μM GDP, and 10 mM 2-mercaptoethanol. The purified protein was concentrated and supplemented with buffered glycerol to 30% final concentration (buffer composition unchanged), then stored at −80. All proteins were checked for purity via SDS-PAGE.

**Free protein and stalled RNC OT samples.** The optical tweezer sample consists of a protein or RNC tethered between two DNA handles. One handle is biotinylated, and through a streptavidin bridge it connects to the biotinylated substrate. The other handle is restriction digested to provide an overhang which is complementary to the chemically cross-linked CoA-oligo group. Both handles have digoxigenin modifications to attach to antidigoxigenin coated polystyrene beads. The DNA handle ligated directly to the sample was generated via PCR of a 2 kb region of the lambda phage genome with a digoxigenin-modified reverse primer and an inserted BsaI site in the forward primer. The primers for the handle generation are listed in Supplementary Table 4. The PCR product was digested with BsaI to produce the 4 nt overhang then gel purified. The second DNA handle was generated via PCR of the same region of the lambda genome with digoxigenin modifications in the reverse primer and biotin in the forward primer, then gel purified.

The coA-oligo was prepared by reacting 40 nmol of amine-modified oligo (IDT, resuspended in water to 1 mM) with 3.75 mmol 4-maleimidobutyric acid N-hydroxysuccinimide ester (GMBS) (50 mM in dimethyl sulphoxide DMSO)) in PBS buffer with 1 mM ethylenediaminetetraacetic acid (EDTA) for 30 min at room temperature. After this time, pelleted material was redissolved in additional DMSO and added back to the reaction supernatant for 15 min to allow for further reaction. This reaction of the amine with GMBS was quenched with excess Tris, ethanol precipitated, and resuspended in 50 mM HEPES with 2.5 mM EDTA to prepare for the next step. In all, 1.25 mmol of coA (97 mM stock in water) was added and reacted for 1 h. The product was gel purified from 7 M urea/15% polyacrylamide, phenol–chloroform extracted, and ethanol precipitated. After resuspension the complementary oligo, which provides the 4 nt overhang, was annealed and the product was stored at −20 or −80.

Released protein samples were CoA-labeled by reacting the purified protein (20 μM) with recombinant Sfp (10 μM) and double-stranded CoA-derivatized oligo (35 μM) in 50 mM HEPES pH 7.5, and 10 mM MgCl2 for 1 h at 30 degrees. This reaction can be stored at −80 for future experiments or further diluted two to tenfold and ligated to the DNA handles. The ligation to DNA handles was conducted in 1× T4 ligase buffer (NEB) with 6 U/uL T4 ligase at 16 degrees for 1 h. The ligated sample was deposited on BSA-passivated antidigoxigenin coated 1 micron polystyrene beads and diluted in 1× polymix (20 mM HEPES, 95 mM KCl, 5 mM MgCl₂, 5 mM NH4Cl, 0.5 mM CaCl₂, 1 mM spermidine, 8 mM putrescine, 0.1 mM DTT, 0.01 mM PMSF, and 0.1 mM benzamidine) supplemented with 10 mM NaN₃ for use in the OT. The biotin handle was deposited on a separate aliquot of BSA-passivated anti-digoxigenin coated 1 micron polystyrene beads, then saturated with a more than 1000-fold excess of streptavidin and also diluted in 1× polymix with azide.

The ribosomes were CoA-modified prior to the stalling reaction by reacting with the CoA-oligo and Sfp in a similar manner as for free protein, then isolating ribosomes through a sucrose cushion[8]. Stalled RNCs were prepared using the PURExpress ΔRibosome kit and adding 0.6 μM-labeled ribosomes, 1 μM birA, 1.5 μM mRNA, 25 μM biotin, 5 mM ATP, and 5 mM Mg(OAc)₂ reacting at 25 degrees for 40 min, then pelleting through a sucrose cushion and resuspending in 1× polymix with 1 U/uL RNAse Out (final concentration of ribosomes is 0.3 μM assuming 100% recovery). All mRNA was designed without a stop codon and generated via run-off in vitro transcription (MEGAscript kit). The transcription template was generated via PCR and gel purified from an agarose TAE gel and ethanol precipitated prior to use. The resulting RNA product was also gel purified from 7 M Urea/4% polyacrylamide gels before use, phenol–chloroform extracted, and then ethanol precipitated before resuspension in 1 mM EDTA pH 8. Stalled RNCs were analyzed for a homogeneous biotinylation product via western blot with a streptavidin-HRP probe before use. After the sucrose cushion, the ligation to DNA handles was conducted in 1× polymix buffer supplemented with 1 mM ATP, 1 mM MgCl₂, 1 mM DTT, and 11 U/uL T4 ligase, then deposited on beads the same as for protein samples and kept on ice.

**Real-time elongation OT samples.** Samples for real-time elongation were stalled via omission of the amino-acid valine. Other amino acids that were not coded before the stall site were also omitted. Stalled ribosomes used for restart were prepared slightly differently than truncated mRNA RNCs owing to the transience of this stalled state and the requirement to omit amino acids. Stalling reactions used 1× concentration of solution A from the PURExpress ΔtRNA, Δaa kit, and 1× concentration of the factor mix from the ΔRibosome kit, as well as 100 μM tRNA, 75 μM biotin, 1 μM birA, 4 μM fMet tRNA, 300 μM amino acid mix (each amino acid is at 300 μM, and the mix contains 15 of the 20 amino acids, omitting arginine, cysteine, proline, tyrosine, and valine), 0.6 μM mRNA and 0.2 μM-labeled ribosomes. Our experiments made via a ³⁵S-Met incorporation timecourse (same concentrations as above except with 25 μM methionine instead of 300 μM and with 2.5 μM ³⁵S-Met) indicate the stalled product is only stable for ~ 10 min before read-through if stalling at a single codon (Supplementary Fig. 5a). Presumably, after this time, the ribosome misincorporates a competing amino acid and continues translating until the end of the sequence. Thus, we used a triple codon repeat which prolongs this stalled state to > 30 min (Supplementary Fig. 5b). We also extended the serine–glycine linker prior to the valine site to ensure that the AviTag was completely outside of the ribosomal exit tunnel and thus accessible for biotinylation and tethering. The inserted sequence was SGSGSGSGSGSGSGSGSGSGSVVV.

To avoid prolonged translation reactions and sucrose cushioning, the reactions were incubated 20 min at 25 degrees. During this time, 2 μL of His-Tag Dynabeads were prepared by washing 2–3 times with 1× polymix + 0.05% (w/v) TWEEN-20. After stalling, the reactions were incubated with His-tag Dynabeads for 10 min on ice to remove the non-ribosomal protein components of the PURE kit (confirmed via silver staining, Supplementary Fig. 5c). The supernatant was removed from the magnetic beads, ligated to the handle for OT experiments in the same manner as for stalled RNCs, and kept on ice until ready to use.

A table of real-time elongation conditions used in the OT is listed in Supplementary Table 3. Plasmids for expression of all 20 tRNA synthetases (RSs) were a gift from Susan Marqusee. Total tRNA (Roche) was pre-charged with S100 and all amino acids, purified via phenol–chlorform extraction and subsequent ethanol precipitation, then resuspended in 1 mM KOAc pH 5.3. These tRNAs were then added to in vitro translation reactions with purified RSs to maintain the equilibrium population of charged tRNAs. Although this step should not be necessary, it removes any lag time while waiting for initial charging to occur and places less stress on the regenerating system since synthetases are ATP dependent. The synthetases then act to recharge any deacylated tRNA, either due to activity of the ribosome or spontaneous deacylation. The required concentration of each synthetase was determined by measuring charging activity with [α³²P]-ATP[47].

**Data collection and processing.** Data were collected on either of two optical tweezer set-ups, both previously described[48,49] and set to the same trap stiffness of 0.25 pN/nm. The traps are generated from 1064 nm lasers focused through high-numerical aperture objectives, allowing us to stably catch particles and measure their position and the force applied. Tethers were formed by catching a bead with sample deposited in one trap and a bead with the biotinylated handle in the other trap, then bringing the beads together and pulling apart until an attachment formed. We used both contour length and curvature of the resulting tethers to ensure single molecules were analyzed.

Data were collected at 2500 Hz for equilibrium measurements and 1333 Hz for elongation measurements, which occur on a longer timescale. Data were used at full frequency for all fittings and analysis, but are displayed downsampled at 250 Hz for equilibrium data figures, 125 Hz for force–extension curves, or 26.67 Hz for real-time data figures, except where indicated. Data collection was done via a LabVIEW custom interface as previously described[48,49]. For data analysis, custom code was written in Matlab R2015b and Python. The HMM was based on open source code in the Python package hmmlearn (v0.2.0).

To convert force data into amino acids, we used a combination of two WLC models. For the DNA, we used an extensible WLC model with persistence length $P = 35$ nm and stretch modulus $S = 1200$ pN[36,50]. These WLC parameters were determined by pulling DNA-only tethers in the same buffer conditions to fit the persistence length and stretch modulus, using a known fixed contour length. The resulting parameters are as should be expected for buffer conditions at high magnesium and polyamine concentrations[51]. For the polypeptide contribution, we used a WLC model with a persistence length of 0.6 nm. A value of 0.6 nm fit our protein tether behavior as well as fit the overall unfolding signature size in force–extension curves, and is similar to previously reported values[24,35]. For converting the contour length (in nm) to amino acids, we used a conversion value of 0.365 nm/aa[24]. These conversions assume the amount of peptide within the tunnel stays approximately constant throughout elongation. Experimentally, values for the number of amino acids within the tunnel vary between 30 and 35 amino acids[14,52]. We used the end point of translation, which has the clear hopping of the C domain and a well-defined length, as an internal reference for the length and back calculated the amino acid addition from the end. In a subset of molecules, we observed occasional small shifts in force in real-time data that we ascribed to inhomogeneity in the flow. These shifts were removed for the conversion to amino acids.

All data were collected in passive mode to avoid artifacts of missed folding transitions and depressed apparent folding rates[53]. In real-time elongation

experiments, one individual amino acid is below our resolution (calculated step size for $+1$ amino acid is $-0.006$ pN at an initial force 4.5 pN and no added amino acids except linkers; calculated step size for $+1$ amino acid to a chain of 224 amino acids at 3.5 pN is $-0.003$ pN). Empirically, our noise at 4 pN is 0.25 pN at 1333 Hz. Assuming a desired signal:noise of 2–2.5, this would be equivalent to 10 steps at 13.33 Hz. Thus, although we cannot say where steps occurred, we can conclude that the chain has elongated by 10 amino acids within a certain window.

Because we cannot identify one amino-acid steps and assigning 10 amino acid steps is unphysical, we instead used a basic isotonic regression which assumes that all the behavior is unidirectional but does not enforce a uniform step size[37,38]. The fitting merely assigns a most probable trajectory—some "steps" will be larger or smaller than one amino acid. Individual steps of the algorithm are not interpreted; the reported translation rate is from the median value of the time to cross a 10-amino acid window, while the error is the standard deviation. For determining τ, the first point of the fit after the specified codon position (codon 167) is used as the time of arrival. The first folding transition is identified by one of two methods. First, the data are divided into 1 sec windows (where the baseline is approximately flat and Gaussian distributed) and downsampled by a factor of 10 in order to better differentiate folding from baseline. owing to this downsampling, our limit of detection will be ~ 8 ms. Each point in the window is compared with a Gaussian CDF with the mean and standard deviation of the window to compute a probability of observing an event of this magnitude or larger. Points which cross a threshold of $1 \times 10^{-5}$ were scored as folding events (corresponding to 99.9% CI) (Supplementary Fig. 6). This scoring method is similar to scoring that has been developed for identifying transitions in force–extension curves, which used a cutoff based on the standard deviation of point-wise difference, where the difference was assumed to be normal[54]. This method scores short events, as long-lived or frequent events can skew the window to be non-Gaussian. This effect can be detected by plotting the standard deviation as a function of time. If the standard deviation passes 0.13, this is also flagged (the second method). This means there is a long-lived event or the molecule has begun hopping very rapidly, for example, C domain hopping. Either of these cases can be identified by eye in the window.

The fit to $\tau_{delay}$ is to the cumulative distribution function (CDF), the integral of the probability distribution function (PDF). The simplest model would correspond to an exponential fit to $1 - e^{-\tau/\tau_{delay}}$. This is a single step stochastic process. We also tested a variety of other potential fits of added complexity: a double exponential distribution and an Erlang distribution. Physically, the double exponential fit would correspond to two populations of ribosomes where one is slowed. An Erlang distribution would correspond to the case where there are two sequential steps of similar rate in the underlying kinetic process. These fits were compared via an $f$ test with the exponential distribution as the null hypothesis as it is the simplest model (Supplementary Fig. 7). The exponential fit has an $R^2 = 0.91$.

Using our fit value to $\tau_{delay}$, our translation rate, and our stalled folding rates, we performed a simulation of ribosome translation. We simulated trajectories of 10,000 ribosomes with and without a delay step prior to folding. Translation times at each codon were drawn from an exponential distribution of 1.0 aa/second. Folding times were drawn from an exponential distribution of $k_{fold} = \frac{1}{1.5}$ seconds$^{-1}$ (from our folding rate at 4.0 pN). As the folding rate is measured as constant between 167 and 177, we maintained a constant folding rate, which was independent of translation. This assumption will likely not hold for the full linker region. The delay step was drawn from an exponential distribution of $\tau_{delay} = 63$ sec. For a simulated CDF, we used the fraction of ribosomes, which had folded at or prior to a given time (or codon position). Translation was stopped at 222 codons, at which point all of the RNCs will fold eventually, leading to a sharp increase in the CDF after codon 221.

**Statistical analysis**. The number of molecules is indicated in the figure legends as "$n = x$ molecules, $y$ rate measurements". Each rate measurement represents an independent fitting of many folding or unfolding events to an exponential cumulative distribution. Error on Bell plots is SE; 95% CIs for the fit parameters are shown in the supplementary tables. Force–extension or passive displays are representative examples of each construct.

One molecule was excluded from folding kinetics fitting of EF123 owing to increased baseline noise that skewed the unfolded lifetime owing to missed folding events to the N domain. The resulting HMM was non-exponential, although EF123 unfolding detection was unaffected and was included. For all protein and stalled RNC data, data were excluded if the total molecular length was outside of the range of 1350–1450 nm at 10 pN, which indicates it may not be the designed single molecule tether (4 kb DNA+sample). For real-time data, molecules were excluded if they did not meet this length and if they did not show C domain folding, which was required in our analysis for calculations. No C domain folding can be owing to no/incomplete translation, tether breakage prior to completion, etc.

Data analysis of equilibrium folding kinetics was done using a HMM python library (hmmlearn) and lifetime fitting in Matlab. Fits were also performed with varying numbers of states, with the best fit selected according to the lowest Bayesian Information Criterion[55]. Rates were obtained by fitting a cumulative density function to fit lifetimes[56]. Because the protein folds and unfolds very quickly and each molecule was tested at multiple forces, one molecule will yield ~ 10 rate measurements from 10 forces, corresponding to 1000 individual folding events. The force data were used for passive analysis with the HMM rather than

extension data owing to offsets observed between bead pairs in extension, likely owing to bead size variations. The trap stiffness and bead offset files owing to trap cross-talk were calibrated for each bead pair after each experiment.

For composite binned data as shown in the Bell plots, the rate data for all molecules for each condition was binned into equally spaced bins along the force axis ($x$ axis). The number of bins was determined by the square root of the number of rate measurements. Then, the linear fit to $\ln(k(F)) = \ln(k_0) - F\Delta x^\dagger/k_B T$ was performed with weighting on the binned data that was $1/\sigma^2$ of each bin or $1/\sigma^2$ from the lifetime fits within the bin, whichever was a greater source of variation (usually the spread of the bin is greater, except in bins at very high or low force, which have more error in the fit and are also less populated). Although more exact models for fitting $\ln(k)$ versus force exist that are non-linear[57], they do not converge well to our data set likely because it only covers a small range of forces (~ 2 pN) such that we do not sufficiently sample the non-linear regions. To test if values from the linear fits ($\ln(k_0)$ or $\Delta x^\dagger$) are equivalent for different experimental conditions, we used a two-tailed Welch's $t$ test for two values with unequal variances. The variances were derived from the fit parameters in Matlab. The number of binned data points in the fit and the variances determine the degrees of freedom according to the Welch-Satterthwaite equation. The null hypothesis is that they are from the same distribution. A $p$ value of < 0.05 indicates they are significantly different values.

## Data availability

Data supporting the findings of this manuscript are available from the corresponding author on reasonable request. A reporting summary for this article is available as a Supplementary Information file. The source data underlying Figs 1c–e, 2c and e, 3a–c, 4a and b and 5a, b, d and e, Supplementary Figs 1, 2, 3, 5, 8 and 9 and Supplementary Tables 1 and 2 are provided as a Source Data file.

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

## Acknowledgements

We thank Professor Ignacio Tinoco Jr., who sadly passed away before this work was completed, for his scientific guidance. We also thank Shannon Yan and Varsha Desai for helpful discussions. We thank Professor Carol Deutsch for editing the manuscript. We thank Apurva Shah and Abigail Keller for assistance in preliminary experiments. This work was supported by the Howard Hughes Medical Institute, by the National Institutes of Health grants R01GM071552 and R01GM032543, and the Nanomachines program (KC1203) funded by the Office of Basic Energy Sciences of the U.S. Department of Energy (DOE) contract no. DE-AC02-05CH11231. L.M.A. was supported by the Department of Defense (DoD) through the National Defense Science & Engineering Graduate Fellowship (NDSEG) Program. L.M.W. is a Fellow of The Jane Coffin Childs Memorial Fund for Medical Research and has been aided by a grant from The Jane Coffin Childs Memorial Fund for Medical Research.

## Author contributions

L.M.A. designed experiments, performed the experiments, analyzed the data, interpreted results, and wrote and edited the manuscript. D.H.G. designed experiments, interpreted results, and edited the manuscript. L.M.W. prepared reagents, designed experiments, and edited the manuscript. C.J.B. designed experiments, interpreted results, and edited the manuscript.
