## [Peer Review File · Nature Communications]

Reviewers' Comments:

Reviewer #1:

Remarks to the Author:

In this study Alexander and co-workers Laser Optical Tweezers used to probe the folding of the protein Calerythrin under three different conditions: off the ribosome, on translationally stalled ribosomes, and on ribosomes undergoing continuous in vitro translation. To monitor folding and misfolding off-the-ribosome or on translationally arrested ribosomes they used force versus extension, and force versus time data (e.g, Figs. 2 and 3). On actively translating ribosomes they transformed their force versus time measurements to contour length versus time, and identified decreases in contour length as transitions to either misfolded or folded states (e.g., Figs. 4 and 5). This studies most important finding is that while misfolding of Calerythrin is observed on stalled ribosomes, it rarely occurs during continuous synthesis. This is a surprising result because the authors observe that when the nascent chain is long enough the misfolded state forms in 1.5 seconds on stalled ribosomes, similar to the average time scale of amino acid addition (1 AA /s) in the expression system. Thus, one would have expected that during continuous translation the misfolded state would be extensively populated, but this does not occur. Based on this, the authors conclude "that comparing equilibrium-derived folding rates from stalled RNCs with translation rates is not sufficient to predict co-translational folding of a polypeptide."

This study is novel and important for several reasons. The co-translational folding field has primarily focused on stalled ribosomes, and it is critical to advance the field through quantitative measurements during continuous synthesis. And as this study demonstrates, probing continuous synthesis provides new, unexpected observations. Furthermore, the observed delay in folding, and suppression of misfolding, is unexpected and has broad implications for studies of co-translational folding as well biological impact.

I do, however, have both technical concerns and concerns over interpretation that should be addressed before I can recommend publication.

Major:

1. From a chemical kinetics perspective, the time evolution of all processes can be accurately described provided (1) you can identify all of the elementary steps in a reaction or conformational transition, and (2) that you can measure all of the rates associated with those steps. You can then stitch this information together through kinetic equations (which describe the time evolution of a collection of molecules), or stochastic dynamics simulations (which describe the time evolution of individual molecules), to predict how the overall system behaves over time. This has been the philosophy of enzymologists for decades: isolate substeps, measure the rates, and then predict overall rates, etc.

In this study, the authors have in some sense taken this approach. They have isolated different points of translation (i.e., different nascent chain lengths) and measured folding and unfolding rates on arrested ribosomes. They have also measured an average elongation rate during continuous synthesis. One thing they did not do is the next step and use this information to predict the time evolution of folding and misfolding populations during continuous synthesis. The authors should do this (i.e., report the predicted probability of folding and misfolding versus time or nascent chain length during continuous synthesis) because it will give readers predicted values to which they can compare the experimental results from continuous synthesis. As part of this analysis, the authors should also report the probability of folding and misfolding measured from the continuous synthesis experiments. (Note that I recognize that the authors did not measure these folding/misfolding rates at every single nascent chain length, and therefore they have sparse data for kinetic modeling. I would find it acceptable, for the purposes of these calculations, to just assume that the rates measured at nascent chain length i are the same as the last measured rate. For example, you measured rates at lengths of 135 and 146 residues; use the rates measured at 135 for lengths 136 through 145.)

2. There is actually a mystery here that the authors do not give much prominence too in their manuscript. Specifically, they have claimed that knowledge of the arrested-ribosome rates of folding/unfolding and codon translation speed are not sufficient to predict what happens during continuous synthesis. In light of my Point 1, above, there are only two hypotheses I can think of that can explain this discrepancy: (i) that the authors were not able to isolate all of the relevant elementary steps necessary to describe continuous synthesis and co-translational folding; or (ii) they could isolate the relevant steps, but weren't able to measure the rates accurately. Given the expertise of the Bustamante group I'm sure hypothesis (ii) is not likely to be true, leaving only hypothesis (i).

In only one sentence in the manuscript do the authors speculate what that other elementary process may be. In the last paragraph of the Results Section they state "This [trapped] state may involve a set of local interactions with the ribosome surface or exit tunnel that were established during its growth in real time or, alternatively, may form only in the context of actively elongating ribosome surface dynamics."

Let's assume that the authors are correct, that there are interactions with the ribosome surface or exit tunnel that influence the folding rate. I would argue that in this case, if the authors were able to isolate and measure the binding and unbinding rates of the protein for the ribosome surface from states U, I, M and F on arrested ribosomes, then they would be able to accurately predict how the protein would behave during continuous synthesis. Do the authors agree with this reasoning? If they do, then they need to remove the claim that "Our results show that comparing equilibrium-derived folding rates from stalled RNCs with translation rates is not sufficient to predict co-translational folding of a polypeptide", and in the Discussion Section they should state that their measurements are missing some crucial processes relevant to continuous synthesis, and that if they were able to isolate those processes on arrested ribosomes and measure their rates they would probably be able to anticipate co-translational folding pathways during continuous synthesis. I do not think such a discussion would detract from the importance and novelty of their overall results. If they disagree with my argument, the authors should be able to explain why chemical kinetic reasoning does not apply to this system.

3. The authors make a crucial claim in their Results section but do not show the data (at least I can't find it in the main text or SI figures.) They state "In support of this interpretation, once the misfolded state forms, ramping the force to unfold it and returning it forces as low as 2 pN, we observe no subsequent delay". I can't verify the accuracy of this claim as they do not show the data; a figure should be inserted showing this.

4. Fig. 4d is concerning to me. There is only a small difference in the peak labelled misfolded, and the peak labelled C domain. This raises the question of how the authors, with such small differences, are able to identify signal from noise, and for a given true signal separate out what is coming from the misfolded state and what is coming from the C-domain. I've read over the Methods Section of the manuscript and it is still not clear to me. The authors should clarify this, and especially whether they apply any statistical tests to differentiate signal from noise.

5. For the same reason as above, Fig. 5 is concerning to me as it is unclear how they label some peaks as misfolded and some as full length, as well as differentiating true folding peaks from noise.

6. Fig. 3a. The meaning of the red lines are not stated in the caption. I presume they are the Markov states. This figure again gives me concern because the M Markov state (the intermediate force between U and F) forces peaks seem similar to the noise in data. Can you provide any controls to demonstrate you are accurately assigning states with your method even in the presence of a large amount of noise?

7. "In a typical force extension curve, at around ~13 pN, full-length calerythrin unfolds...". Since these are stochastic systems with an underlying distribution the authors should insert the distribution of ripping forces in the SI so readers can assess how representative 13 pN is.

8. Confusion over phrasing: "The passive data shows hopping between unfolding and an intermediate state before full-length folding, which is long-lived." What is long lived? The folded state or the intermediate state?

9. Confusion over phrasing: "...and tethered them via the N terminus of the nascent chain and a tag on protein L17 of the ribosome." What elements are being tethered together? Literally read one could interpret this sentence to mean the N-terminus and protein L17 are being tethered together. I assume the original sentence means each is tethered to a DNA handle?

10. Why do the authors switch the visual representation of their data from force versus time (for off-ribosome and stalled ribosome systems) to contour length versus time for continuous synthesis? It makes it harder to compare and contrast these three systems. To help with such a comparison it would be ideal if the authors could be consistent across all systems, but if that is not appropriate, at a minimum I think they should show in the SI the Contour Length versus time data for the off-the-ribosome and stalled ribosome systems in which the contour length jumps between U, I, M and F are seen.

Minor:

1. "(Supplementary Fig.3)", space needed between Fig. and the number 3.
2. Methodologies should have references upon first use, for example, provide relevant citations for WLC and isotonic regression analyses.
3. "stall complexes" I assume should read "stalled complexes"
4. I think it is questionable whether Fig. 5d clearly demonstrates an exponential distribution as no statistical tests for alternative distributions are provided. Either provide those tests or weaken the claim that "The distribution of measured τ values is exponential"
5. I don't know what the reporting requirements are for statistics in Nature Communications, but I think it is best practice to state the following information: Statistical test name, n=sample size; p-value=X. For example, the authors should do this in-line for "... lower forces than FL_int (p<0.05)"

Reviewer #2:

Remarks to the Author:

Alexander et al have used single-molecule optical trap (OT) measurements to characterize the folding/unfolding transition of a two-domain Ca-binding protein, (1) in vitro, (2) when stalled on a ribosome, and (3) during ongoing translation.

Under condition 1, the protein folds through an intermediate that corresponds to the folding of the C-domain.

Under condition 2, when only the N domain is outside the ribosome no folding is observed. When the N domain and part of the C domain are outside the ribosome the protein folds into a misfolded state that may represent a domain-swap between the N and C domains. Finally, when the entire protein is outside the ribosome, the protein folds through the C domain, as it does in vitro. The ribosome decelerates folding and accelerates unfolding of the misfolded state by orders of magnitude.

While the behavior under conditions 1 and 2 are as expected from previous single-molecule studies of folding on the ribosome, condition 3 yields a novel result, not observed before: the onset of folding to the misfolded state or to the folded C domain is delayed to long nascent chain (NC) lengths, much longer than expected from the experiments with the stalled ribosomes. Under in vitro translation conditions, this delay corresponds to the synthesis of approximately 70 additional

residues beyond the point when the whole protein is expected to be outside the ribosome. This is interpreted to mean that the NC is produced on the ribosome in a trapped, folding-incompetent state that only slowly makes the transition into a folding-competent state. Although the nature of this hypothetical folding-incompetent state is completely obscure, this observation is of course intriguing. It leaves one wondering what is really going on, though.

Major comments:

1. From the trace in Figure 4, it seems that the ribosome reaches the end of the truncated mRNA well before folding starts; i.e., it reaches the stalled state first, and only then does the NC start to fold. Is this correct? For what fraction of the $n=12$ molecules observed is this the case? As far as I understand, the NC length from residue 167 to the end of the polypeptide is $(177-167)+45 = 55$ amino acids and the mean delay time is 71 ± 14 s. With a measured translation rate of 1.0 ± 0.3 aa/s, one would expect the synthesis of 55 amino acids to take 55 ± 17 s, which would be consistent with nearly all NCs reaching the stalled state before folding is seen to begin. If this is indeed the case, what one can conclude is that, under condition 3, most NCs do not fold during ongoing translation, but only from the stalled state, as under condition 2. If this is correct, it makes one wonder if there is something about the assay itself that prevents folding of the NC during translation.
2. One obvious difference between conditions 2 and 3 is that under the latter the NC in the ribosome-nascent chain complex is under constant tension, being pulled on by a force between 3.5-4.5 pN at all times. This is a non-physiological situation, and I wonder whether this in itself could prevent folding of the NC during translation (perhaps in conjunction with the force that ratcheting of the ribosome against the mRNA might produce on the NC)? Can an experiment be designed where translation is initiated at $t=0$, but pulling force is applied only later at a time that correspond to, e.g., the emergence of the first 167 residues from the ribosome? Would any (mis)folded molecules be apparent under such conditions?
4. The authors propose no mechanistic explanation for the postulated folding-incompetent state of the growing NC – this is an obvious weakness of the paper.

Minor comments:

4. Please explain the difference between measurements in “active” and “passive” modes, as a help to readers not expert in single-molecule OT.
5. In the Discussion, it is mentioned that no folding delay was seen in the real-time FRET analysis of cotranslational folding of the HemK protein (ref 13), and this is ascribed to poor synchronization in the FRET assay. The FRET assay is synchronized, however (ref 18), and a massive delay in folding such as that reported here ought to be clearly visible in the data. Of course, the HemK protein is not the same as the Ca-binding protein analyzed here, but I don't think that the FRET results can be discounted on the basis of poor synchronization. In the HemK case the NC is not being pulled on during translation, and hence not under tension.
5. In Methods, it is stated that the real-time elongation experiments were done in an amino acid mix with “all 20 amino acids except arginine, cysteine, proline, tyrosine, or valine”. Does this mean that none of the amino acids were included, or that different experiments excluding different amino acids were done? Only valine exclusion is mentioned in the Ms as a way to synchronize the reaction; what were the results when omitting the other ones?
6. Figures 2-5: Please indicate the different folding/refolding events mentioned in the legends by arrows.
7. Figure 3a: Please explain in the legend what is different between the red and black traces.
8. Figure 4d: I understand that folding into the misfolded state and into the folded C domain yield peaks of different magnitudes. It would be good to have a simple shorthand explanation for how reliably one can distinguish between the two kinds of events (and distinguish them from other events such as the somewhat smaller peaks seen upstream of the “misfolded” peak in the trace). If there are more peaks that you have assigned as “misfolded” or “C domain” in this trace, please indicate them also, just to help the uninformed reader get a better feel for the assay.
9. Supplementary Fig. 4. In what way do the linkers differ? Please include the full sequence of the normal constructs as well as the ones with altered linker (I assume that it's the linker in the protein that differs, not the DNA handles?). Also, please detail the conditions $\pm TM$ in the legend.

Reviewer #3:

Remarks to the Author:

Alexander et al. use a single-molecule translation assay to study how active translation by the ribosome may affect the folding of nascent polypeptides. By comparing the folding of a given protein after translation (without ribosomes present), during arrested translation, and during active translation, they find that there are noticeable differences (most notably, delayed folding during active translation), which suggest that the folding is not in equilibrium during translation. Furthermore, this work shows that the co-translational folding kinetics may differ from what would be expected based on studies of folding off the ribosome as well as on stalled ribosomes.

This manuscript addresses an important question, one that has proven difficult to study using traditional methods, and it contains some very interesting and beautiful results. It should be of interest to the broad readership of Nature Communications, and will be suitable for publication after addressing the concerns noted below.

1. Page 3, first paragraph: when discussing previous work on EF-hand family proteins, it would be a good idea to include a reference to Heidarsson et al PNAS 2014, which discusses Ca-dependent misfolding (of direct relevance to this work).

2. Page 3: when discussing the pulling-curve results, it would be a good idea to mention that the length change matched what was expected for the native structure.

3. Page 3: the glutamine mutation in EF1 doesn't affect formation of FLint, but it does seem to change the unfolding from two-state by introducing an intermediate. This difference is worth mentioning.

4. Page 4, top: the basic results that the ribosome destabilizes structures near the N-terminal domain in the nascent chain, and that overall folding is slower with the ribosome present, are not really new, as Kaiser et al saw exactly the same thing previously in their work comparing folding off and on the ribosome. But the extent of the changes seems to be different here, since the destabilisation is enough to prevent folding entirely. These connections to previous work (and their significance) should be discussed explicitly.

5. Page 4, paragraph 2: the call-out should be to Table S2, not Table S1.

6. Page 4, paragraph 3 and following: Although most of the identifications of particular states with proposed structures in this manuscript seem well-founded, the interpretation of some of the states in the stalled complexes as misfolded is not convincing. (To be clear: the interpretation is not unreasonable, and not necessarily wrong, it's just not proven properly.) A misfolded state is typically defined as a stable or metastable state that is off the native pathway--i.e. there is a large enough barrier for transitions between that state and the native state that direct transitions from the misfolded state to the native state do not happen (otherwise, the putative 'misfolded' state would actually just be part of an alternative pathway to the native state). The problem here is that the authors have used the length change to deduce that a state is not native, but length changes alone may not be sufficient to establish that a state is misfolded. Other explanations are possible and must be ruled out, for example it could be that the state is actually an intermediate on the native pathway that is not observed under normal circumstances but becomes detectable because of the truncation.

The authors claim that the 'misfolded' state is indeed off-pathway because it never transitions to the native NTD fold, but they provide no analysis proving this claim. Indeed, even without access to the full dataset, close inspection of the data presented in Fig. 3 suggests that the claim that

there are no transitions from the 'misfolded' state to the native NTD fold is incorrect: one can see what looks like a very brief pause in the folded NTD state as the 'misfolded' state transitions to unfolded in a brief event at ~ 0.8 or 0.9 s in the record. One can also see very brief "spikes" down from the 'misfolded' state to a level consistent with the folded NTD at ~ 1 s and ~ 1.4 s. The short lifetime of the folded NTD state may be confounding the analysis by making it more difficult to see such transient intermediates in the folding/unfolding of the putatively 'misfolded' state (certainly, given the relatively low sampling rate of the data, one would not necessarily expect to see such a transient intermediate on every transition into/out of the 'misfolded' state).

Given the importance of this putative misfolded state to the authors' interpretation, more careful analysis of this state is needed. The authors should examine their data more carefully for signs that there are very brief intermediates during the formation of the 'misfolded' state (on transitions to/from unfolded) or during fluctuations out of the 'misfolded' state. The methods described in Yu et al PNAS 2012 for detecting transient misfolded states could be useful for this analysis. More centrally, the authors need to include some analysis of whether this putative 'misfolded' state can or cannot transition directly into the native fold, since that is the true indication that it is misfolded. Even if the 'misfolded' state forms after the NTD folds natively, it could still end up being an off-pathway misfolded state (as in the example in calmodulin observed by Stigler et al).

7. Page 5, top: The extrapolation of the 'misfolded' state occupancy at zero force seems suspect. Static measurements don't really provide a good basis for projecting the relative (equilibrium) occupancies during elongation because the relative stability of the different states (and the rates for leaving this partially folded state) may very well depend on the length of the polypeptide chain and hence vary with time. Given that the authors describe active ribosome measurements in the next section, this crude and somewhat misleading estimate does not seem to be needed.

8. Page 5, paragraph 3: In figure 4 c and d, it is unclear how states are being detected and assigned. The experimental errors/uncertainties look much larger than in the static ribosome measurements, for understandable reasons. The claim that the N state (NTD folded) doesn't occur is not convincing from these data. For example, in Fig. 4d there are "spikes" upward like at ~ 274 s that look like the right size for folding of N (smaller than C, as expected from the previous data). Also, are the authors assuming that even though the kinetics have apparently changed a lot from the stalled ribosome data, there are no additional (different) states formed here? More robust analysis of the data to back up the claims (or appropriate modifications of the claims to match the evidence presented) is needed.

9. Page 5, bottom: if state N is present, presumably the analysis of the pathways would have to be changed.

10. Page 5, bottom: regarding the delay times τ , the authors claim that an exponential distribution (leading to a curved CDF) fits the data well, but in fact the CDF is very clearly linear--the fit is systematically off. The authors claim a very good fit quality (R^2), but the problem is that the deviations are systematic, not random, which indicates that the model is not good (it's very likely that a test such as Wald-Wolfowitz should show that the residuals to the fit are not actually random). It therefore seems that a non-exponential distribution of times is present and a different model should be used to fit the data (perhaps something like a stretched exponential?). This non-exponential distribution is actually quite interesting, as it has important implications for the origin of the delay. Certainly, the interpretation presented here that the equilibration happens via a single stochastic step (giving rise to exponential time distribution) is not well supported by the data, and a more complex model is needed. Is there some strong reason to expect that the delay *should* be exponential? If so, what are the implications of the fact that it clearly is not? This issue should be explored further in a revised manuscript.

11. Page 5 bottom/6 top: The data in Fig 5 clearly show brief "spikes" of transient folding before the delay time identified on the figure is finished. For example, in Fig 5a there is a spike at ~ 225 s,

in 5b there is one at ~210 s, in 5c there is one at ~110 s. They look like possibly folding of N or C, but it's unclear; they just don't lead to anything stable. The events seem to be ignored entirely by the authors? They need to be discussed and incorporated into the interpretation!

12. Page 7, top: If the "unfolded" state has a much longer lifetime in the presence of the active ribosome, then clearly it can't truly be the same "unfolded" state as in the absence of the ribosome--there must be some interaction or process that differentiates it with the ribosome present and slows down the folding.

13. Page 7: These experiments provide a very nice way to probe how the dynamic ribosome influences the folding, but of course they don't recapitulate the situation in the cell (e.g. lack of crowding, which could have an important effect on the relative stability of partially folded and/or misfolded states). Some brief acknowledgement of this issue would be helpful in the Discussion section.

14. Page 10, end of first paragraph: the silver-staining data should be added to SI.

15. Page 10, paragraph 4: please clarify which parameters were treated as fixed and which were used as fitting variables. The wording seems to suggest that for the DNA handles, the persistence length and stretch modulus were fixed but the contour length was fitted, which seems backwards in some sense (the contour length is the one parameter that is known precisely for the handles).

16. For all the figures containing pulling curves, please show the worm-like chain (WLC) fits to each different state, to help assess the quality of the data. Also, please add some of the data at low force so that the 'knee' of the force-extension curve (which is most reflective of persistence length) can be seen.

17. In the Methods section, please describe how it was ensured that only single molecules were being measured. On page 11 it is implied that the contour length alone is used, but that is known to be an imperfect indicator of single vs multiple tethers between the beads.

18. Page 12, top: please clarify what property is calibrated after each experiment.

19. Page 12, end of Methods: there is growing acknowledgement that the criterion $p < 0.05$ is too weak to be reliable, as it yields many false positives (see e.g. Calquhoun 2014). The authors would be advised to treat results close to $p \sim 0.05$ (as for one of the results in Table S1) with skepticism as to whether it is truly significant.

20. Figure 2C: Please clarify if these curves are unfolding or refolding. Note that there are clearly some dynamics going on in construct 167, but they are not mentioned/discussed clearly in the manuscript. WLC fits should be added to ascertain if the protein is purely unfolded as claimed, or if there are rapid or non-cooperative transitions present (which can be quite subtle, producing curves that 'slide' from one WLC to another as seen in Zoldak PNAS 2013 or Solanki PRL 2014).

21. Figure 2D: the color- and dash-coding of the curves are difficult to understand. Perhaps a rainbow color-coding could be used?

22. Fig 3A: Please add dotted lines showing the position expected for the intermediate state proposed here. Fig 3B: Please clarify whether these pulls are unfolding or refolding. Also, the unfolded states appear to be slightly misaligned. Fig 3C: please explain the figure symbols in the caption.

23. Fig S1: The caption in (B) says the fully folded state is not stable, but this seems incorrect since it forms when the force is ramped down. Please clarify or correct.

24. Fig. S1: For all the mutants, please list the measured length changes and how they compare to the expected length changes (either in the caption or in a table), and of course show the WLC fits.
25. Fig S2: Since the force is varying throughout these measurements, please clarify what is meant by "at 3.5 pN".
26. In the SI, please add a figure showing the distributions of lifetimes for each state, and any exponential fits to them, so that the readers can assess the data used to evaluate the kinetic rates.
27. Tables S1, S2: rather than showing the "size" of the states in amino acids, which is an inferred quantity, it would be better to report the measured quantity, length change, and then compare to the length change expected from the proposed structures. Please describe in the Methods how the conversion between length changes and amino acids is done.
28. Tables S1, S2: the errors in the rates do not seem to be treated properly. Given that they are obtained from a linear fit to $\ln(k)$, the rates should be reported in exponential notation with the errors in the exponent. This approach should get rid of the "unphysical" errors the authors mention that they obtained in some cases.
29. The authors may wish to expand their discussion by outlining why calerythrin differs from calmodulin folding (in terms of intermediates) as found previously by Stigler et al.

Reviewer #4:

Remarks to the Author:

The manuscript describes the folding of the nascent polypeptide chain in and outside the ribosome tunnel. The object for this research is well chosen, the multi-domain protein calerythrin from bacterium *S.erythraea*. It consists of four EF hands, two in the N domain and two in the C domain. The paper describes the interaction between the rate of synthesis and the folding for the whole protein and the two domains. All this is observed in single molecule experiments using optical tweezers.

The manuscript is carefully written providing good evidence for the conclusions. The results are new showing the interaction of peptide synthesis and folding in time. In this respect the paper is showing important details of folding and misfolding of the whole protein and the N and C terminal domains. The understanding of protein synthesis and folding is of principal interest in cell biology. The manuscript can be published as it is. However, it should be clearly stated that this in vitro system does not contain the ribosomal trigger factor or other chaperons like Dnak or GroEL. On the other hand it would have been quite interesting to have seen how things are changing in the presence of these chaperons. The authors mentioned that they observed misfolding. What is the structure of these misfolded nascent polypeptides? Maybe they could shortly mention interesting NMR investigations of C.M. Dobson et al. who found that an initial N terminal nascent chain showed a β -strand which converted to a helical structure when the chain became longer. Maybe the authors have evidence about the type of structure of these misfolded polypeptide chains.

Reviewers' comments:

Reviewer #1 (Remarks to the Author):

In this study Alexander and co-workers Laser Optical Tweezers used to probe the folding of the protein Calerythrin under three different conditions: off the ribosome, on translationally stalled ribosomes, and on ribosomes undergoing continuous in vitro translation. To monitor folding and misfolding off-the-ribosome or on translationally arrested ribosomes they used force versus extension, and force versus time data (e.g, Figs. 2 and 3). On actively translating ribosomes they transformed their force versus time measurements to contour length versus time, and identified decreases in contour length as transitions to either misfolded or folded states (e.g., Figs. 4 and 5). This studies most important finding is that while misfolding of Calerythrin is observed on stalled ribosomes, it rarely occurs during continuous synthesis. This is a surprising result because the authors observe that when the nascent chain is long enough the misfolded state forms in 1.5 seconds on stalled ribosomes, similar to the average time scale of amino acid addition (1 AA /s) in the expression system. Thus, one would have expected that during continuous translation the misfolded state would be extensively populated, but this does not occur. Based on this, the authors conclude “that comparing equilibrium-derived folding rates from stalled RNCs with translation rates is not sufficient to predict co-translational folding of a polypeptide.”

This study is novel and important for several reasons. The co-translational folding field has primarily focused on stalled ribosomes, and it is critical to advance the field through quantitative measurements during continuous synthesis. And as this study demonstrates, probing continuous synthesis provides new, unexpected observations. Furthermore, the observed delay in folding, and suppression of misfolding, is unexpected and has broad implications for studies of co-translational folding as well biological impact.

I do, however, have both technical concerns and concerns over interpretation that should be addressed before I can recommend publication.

Major:

1. From a chemical kinetics perspective, the time evolution of all processes can be accurately described provided (1) you can identify all of the elementary steps in a reaction or conformational transition, and (2) that you can measure all of the rates associated with those steps. You can then stitch this information together through kinetic equations (which describe the time evolution of a collection of molecules), or stochastic dynamics simulations (which describe the time evolution of individual molecules), to predict how the overall system behaves over time. This has been the philosophy of enzymologists for decades: isolate substeps, measure the rates, and then predict overall rates, etc.

In this study, the authors have in some sense taken this approach. They have isolated different points of translation (i.e., different nascent chain lengths) and measured folding and unfolding rates on arrested ribosomes. They have also measured an average elongation rate during continuous synthesis. One thing they did not do is the next step and use this information to predict the time evolution of folding and misfolding populations during continuous synthesis. The

authors should do this (i.e., report the predicted probability of folding and misfolding versus time or nascent chain length during continuous synthesis) because it will give readers predicted values to which they can compare the experimental results from continuous synthesis. As part of this analysis, the authors should also report the probability of folding and misfolding measured from the continuous synthesis experiments. (Note that I recognize that the authors did not measure these folding/misfolding rates at every single nascent chain length, and therefore they have sparse data for kinetic modeling. I would find it acceptable, for the purposes of these calculations, to just assume that the rates measured at nascent chain length i are the same as the last measured rate. For example, you measured rates at lengths of 135 and 146 residues; use the rates measured at 135 for lengths 136 through 145.)

We agree with the reviewer that we can use our measured rates to predict the time evolution of the system, assuming all of the substates have been identified. We have in fact done this in the time domain, where we show the expected cumulative distribution function (CDF) CDF for time until first folding (Figure 5d, black dotted line). We did not explicitly address the evolution in sequence-space, generalizing to “shortly after a minimum length is synthesized”. We agree with the reviewer that this generalization can be more quantitatively addressed through modeling the kinetic events. Therefore, we have performed the time evolution relative to codon position and show the fraction of molecules that are predicted to fold by the time translation reaches a certain codon position (in this case we ignore the unfolding rate and model the folded state as an absorbing state, since we are only interested in when the first transition occurs). This prediction is added as a panel to Figure 5. If the real-time misfolding trajectories rates would match those of the stalled complexes, 99.6% of RNCs would misfold by codon 177 (black dotted lines in Figure 5e). Instead, in our experiment (blue dots) only 16.7% of ribosomes misfold by this point, and 42% of ribosomes misfold after codon 217, within five codons of the end of our mRNA. Note that the empirical CDF is skewed because the sequence ends at 222 (177+45 linker), so all of the ribosomes must stall and misfold at 222, given enough time, rather than continue translating, hence the CDF does not gradually but sharply approaches 1. We have also performed the kinetic modeling with an added exponentially distributed delay time of 71 seconds prior to folding and have added this to our data in sequence space (red dotted line).

2. There is actually a mystery here that the authors do not give much prominence too in their manuscript. Specifically, they have claimed that knowledge of the arrested-ribosome rates of folding/unfolding and codon translation speed are not sufficient to predict what happens during continuous synthesis. In light of my Point 1, above, there are only two hypotheses I can think of that can explain this discrepancy: (i) that the authors were not able to isolate all of the relevant elementary steps necessary to describe continuous synthesis and co-translational folding; or (ii) they could isolate the relevant steps, but weren't able to measure the rates accurately. Given the expertise of the Bustamante group I'm sure hypothesis (ii) is not likely to be true, leaving only hypothesis (i).

In only one sentence in the manuscript do the authors speculate what that other elementary process may be. In the last paragraph of the Results Section they state “This [trapped] state may involve a set of local interactions with the ribosome surface or exit tunnel that were established during its growth in real time or, alternatively, may form only in the context of actively elongating ribosome surface dynamics.”

Let's assume that the authors are correct, that there are interactions with the ribosome surface or exit tunnel that influence the folding rate. I would argue that in this case, if the authors were able to isolate and measure the binding and unbinding rates of the protein for the ribosome surface from states U, I, M and F on arrested ribosomes, then they would be able to accurately predict how the protein would behave during continuous synthesis. Do the authors agree with this reasoning? If they do, then they need to remove the claim that "Our results show that comparing equilibrium-derived folding rates from stalled RNCs with translation rates is not sufficient to predict co-translational folding of a polypeptide", and in the Discussion Section they should state that their measurements are missing some crucial processes relevant to continuous synthesis, and that if they were able to isolate those processes on arrested ribosomes and measure their rates they would probably be able to anticipate co-translational folding pathways during continuous synthesis. I do not think such a discussion would detract from the importance and novelty of their overall results. If they disagree with my argument, the authors should be able to explain why chemical kinetic reasoning does not apply to this system.

Our point in making the claim in the discussion is that from just the equilibrium derived folding rates, it may not be possible to predict the real-time folding behavior of the chain because some elementary substeps present in the system may not be evident or easily accessible in equilibrium measurements. In general, nonetheless, we agree with the reviewer that if all the elementary steps are measured then we should be able to predict the cotranslational behavior at equilibrium. Therefore, we believe that chemical kinetics reasoning *does* apply to this system. However, in practice, it is not always feasible to follow the reaction (folding, misfolding) as a function of time, since preparation of samples, tethering them between beads, and analyzing the molecular state (folded, misfolded) can hardly be done in less than 30 minutes. Now, we realize that this is a technical rather than a conceptual difficulty, but it is nonetheless real and our point is that all these difficulties are overcome if one can follow in real-time the dynamics of the system. We note that this limitation in the study of stall complexes is not exclusive to single molecule optical tweezers experiments. In any experiment in which stalled complexes are used, enough time must be allowed for the entire population to complete the reaction and this requirement often involves waiting long enough times.

In the manuscript, we claim that “*Our results show that comparing equilibrium-derived folding rates from stalled RNCs with translation rates is not sufficient to predict co-translational folding of a polypeptide*” By “equilibrium” we meant the situation described above wherein one must wait a period of ~ 30 minutes to make an observation. It is in this sense that we make that statement in the text. We have now added a section in the manuscript where we clarify the practical limitations of a full kinetic analysis using stalled complexes and the advantages of using real-time measurements. This part of the manuscript now reads:

“Stalled complexes require, for most experimental techniques, at minimum tens of minutes to prepare, which allows time for equilibration processes to occur. If these processes could be induced in stalled complexes, we would expect to be able to predict the co-translational behavior; however, since the exact nature of this process is unknown and was not previously predicted, the measurement must be conducted in real-time.”

3. The authors make a crucial claim in their Results section but do not show the data (at least I can't find it in the main text or SI figures.) They state “In support of this interpretation, once the

misfolded state forms, ramping the force to unfold it and returning it forces as low as 2 pN, we observe no subsequent delay". I can't verify the accuracy of this claim as they do not show the data; a figure should be inserted showing this.

We thank the reviewer for pointing to us this omission. Thus, we have now added some example molecules to the supplemental figures (Supplemental Fig. 9) that show that the long waiting period (delay) before hopping is observed, does not re-occur once the molecule is subjected to a force-ramp.

4. Fig. 4d is concerning to me. There is only a small difference in the peak labelled misfolded, and the peak labelled C domain. This raises the question of how the authors, with such small differences, are able to identify signal from noise, and for a given true signal separate out what is coming from the misfolded state and what is coming from the C-domain. I've read over the Methods Section of the manuscript and it is still not clear to me. The authors should clarify this, and especially whether they apply any statistical tests to differentiate signal from noise.

In the case of real-time figures (Figures 4 and 5), the data shown is heavily down sampled in order to display on a reasonable timeframe (~ 300 s in Figure 4b and ~ 40 s in Figure 4d). Thus the bandwidth of the real-time data is 27 Hz and in this scale the visit to a state appears as a single spike resembling a noise event. We have revised figure 4d so that rather than showing ~45 seconds of data, it displays now 2 seconds of data which is displayed at higher frequency (267 Hz). As shown in the new figure, it is possible to observe the residence time of the molecule in a misfolded state. We have also shown an additional inset, 4e, of the hopping behavior to both C and M when translation has progressed further in the template. We have also overlaid the expected transition sizes for the C and M states to help guide the reader.

For stalled RNC data, transitions are identified via the HMM fitting. In this case the model makes an assignment based on the highest likelihood. Because the HMM is in itself a statistical test, we do not perform additional tests to determine individual transitions from noise, although we do perform statistical tests against different outputs of the HMM (different initialization conditions) using the BIC (Bayes Information Criterion). The quality of the HMM can be judged by the CDFs that are generated. We have added sample CDFs for different constructs so the quality can be judged (Supplementary Figure 1).

Additionally, we have added a section on identifying transitions in the real-time data (Supplementary Fig. 6). Due to concerns about distinguishing events from noise or missed events, we have modified our detection limit to ~8 ms (previously we used 20 ms), which accounts for the change in our reported delay from 71 seconds to 63 seconds.

5. For the same reason as above, Fig. 5 is concerning to me as it is unclear how they label some peaks as misfolded and some as full length, as well as differentiating true folding peaks from noise.

In Figure 5 we wished to display the observed delay time, thus requiring us to show a long time series, where the visit to a discrete state appears as a spike. In light of the reviewer's comment, we have provided additional insets to each panel where the data are zoomed to the folding regions and shown at higher frequency. We have updated the caption to reflect this change and emphasize the display frequency.

6. Fig. 3a. The meaning of the red lines are not stated in the caption. I presume they are the markov states. This figure again gives me concern because the M Markov state (the intermediate force between U and F) forces peaks seem similar to the noise in data. Can you provide any controls to demonstrate you are accurately assigning states with you method even in the presence of a large amount of noise?

In Figure 3a, we have now labeled states U, N, and M for clarity. The state the reviewer is concerned about is state N, rather than state M. The caption now also identifies the red line as corresponding to the HMM. We have also added a supplemental figure (new Supplementary Fig. 4), which confirms that the N state is a true state.

In general, rather than make strict cutoffs for state detection and noise, which is prone to miss events, we use the convergence of the HMM with well-defined, exponential lifetimes to verify the number of states. In the case of fitting arbitrary states to noise, the lifetimes will either not be exponential or they will not have force dependence or both.

7. “In a typical force extension curve, at around ~ 13 pN, full-length calerythrin unfolds...”. Since these are stochastic systems with an underlying distribution the authors should insert the distribution of ripping forces in the SI so readers can assess how representative 13 pN is.

We agree that the unfolding force is a stochastic value and information should be provided about its spread. We have instead elected to report the standard deviation of the unfolding force in the main text. We are happy to add the distribution to the SI if necessary. This now reads:

“In a typical force-extension curve, at around 13 ± 1.3 pN (s.d.), full-length calerythrin unfolds in a single rip indicating cooperative unfolding (Fig. 1c).”

8. Confusion over phrasing: “The passive data shows hopping between unfolding and an intermediate state before full-length folding, which is long-lived.” What is long lived? The folded state or the intermediate state?

We have clarified that the folded state is long-lived by separating this sentence in to two. It now reads:

“The passive data shows hopping between unfolding and an intermediate state before full-length folding (Fig. 1d). Full-length folding is long-lived at this force and must be unfolded at high force to observe further transitions.”

9. Confusion over phrasing: “...and tethered them via the N terminus of the nascent chain and a tag on protein L17 of the ribosome.” What elements are being tethered together? Literally read one could interpret this sentence to mean the N-terminus and protein L17 are being tethered together. I assume the original sentence means each is tethered to a DNA handle?

We have clarified the tethering geometry of the RNCs in the text. We also updated the description of the tethering geometry of the full-length protein to reflect the same wording. This now reads:

“We prepared RNCs stalled at codons 135 to 177 in 10–11 codon increments (Fig. 2a) and tethered them via the N terminus of the nascent chain to one DNA handle and a ybbR tag on protein L17 of the ribosome to the other DNA handle (Fig. 2b)⁸.”

10. Why do the authors switch the visual representation of their data from force versus time (for off-ribosome and stalled ribosome systems) to contour length versus time for continuous synthesis? It makes it harder to compare and contrast these three systems. To help with such a comparison it would be ideal if the authors could be consistent across all systems, but if that is not appropriate, at a minimum I think they should show in the SI the Contour Length versus time data for the off-the-ribosome and stalled ribosome systems in which the contour length jumps between U, I, M and F are seen.

For continuous synthesis, we showed CL vs time in order to show when the ribosome reaches codon 167 and is, in principle, able to fold. We then used the average force at this point to compare to the kinetics gathered initially. We can certainly show contour length vs. time for all data. However, the relevant independent variable would be force, since the contour length of the transition is independent of force. We recognize that this change in plotting can be confusing for the reader and have switched the real-time to force vs time to be consistent with earlier figures as the reviewer suggested, marking codon 167 with a dashed blue line.

Minor:

1. "(Supplementary Fig.3)", space needed between Fig. and the number 3.
2. Methodologies should have references upon first use, for example, provide relevant citations for WLC and isotonic regression analyses.
3. "stall complexes" I assume should read "stalled complexes"
4. I think it is questionable whether Fig. 5d clearly demonstrates an exponential distribution as no statistical tests for alternative distributions are provided. Either provide those tests or weaken the claim that "The distribution of measured τ values is exponential"

We have addressed the minor concerns noted and appreciate the careful reading of the manuscript. We have added additional citations to the main text and the methods. We have provided tests for an exponential distribution (one dominant kinetic step) vs. a double exponential (parallel steps, i.e. a subset of ribosomes are slowed) and an Erlang distribution (two steps in series of similar rate). These fits are compared via an f-test and are described in the methods, as well as shown in Supplementary Fig. 7.

5. I don't know what the reporting requirements are for statistics in Nature Communications, but I think it is best practice to state the following information: Statistical test name, n =sample size; p -value= X . For example, the authors should do this in-line for "... lower forces than FL_int ($p < 0.05$)"

Exact p values for all tests are reported in the Supplementary Tables. The sample sizes are reported in captions for figures. In the case the reviewer mentions, we have added the statistical test to the main text, but we are comparing p values for 4 different parameters between FL and N domain, so reporting in line may be confusing.

Reviewer #2 (Remarks to the Author):

Alexander et al have used single-molecule optical trap (OT) measurements to characterize the folding/unfolding transition of a two-domain Ca-binding protein, (1) in vitro, (2) when stalled on a ribosome, and (3) during ongoing translation.

Under condition 1, the protein folds through an intermediate that corresponds to the folding of the C-domain.

Under condition 2, when only the N domain is outside the ribosome no folding is observed. When the N domain and part of the C domain are outside the ribosome the protein folds into a misfolded state that may represent a domain-swap between the N and C domains. Finally, when the entire protein is outside the ribosome, the protein folds through the C domain, as it does in vitro. The ribosome decelerates folding and accelerates unfolding of the misfolded state by orders of magnitude.

While the behavior under conditions 1 and 2 are as expected from previous single-molecule studies of folding on the ribosome, condition 3 yields a novel result, not observed before: the onset of folding to the misfolded state or to the folded C domain is delayed to long nascent chain (NC) lengths, much longer than expected from the experiments with the stalled ribosomes. Under in vitro translation conditions, this delay corresponds to the synthesis of approximately 70 additional residues beyond the point when the whole protein is expected to be outside the ribosome. This is interpreted to mean that the NC is produced on the ribosome in a trapped, folding-incompetent state that only slowly makes the transition into a folding-competent state. Although the nature of this hypothetical folding-incompetent state is completely obscure, this observation is of course intriguing. It leaves one wondering what is really going on, though.

Major comments:

1. From the trace in Figure 4, it seems that the ribosome reaches the end of the truncated mRNA well before folding starts; i.e., it reaches the stalled state first, and only then does the NC start to fold. Is this correct? For what fraction of the $n=12$ molecules observed is this the case? As far as I understand, the NC length from residue 167 to the end of the polypeptide is $(177-167)+45 = 55$ amino acids and the mean delay time is 71 ± 14 s. With a measured translation rate of 1.0 ± 0.3 aa/s, one would expect the synthesis of 55 amino acids to take 55 ± 17 s, which would be consistent with nearly all NCs reaching the stalled state before folding is seen to begin. If this is indeed the case, what one can conclude is that, under condition 3, most NCs do not fold during ongoing translation, but only from the stalled state, as under condition 2. If this is correct, it makes one wonder if there is something about the assay itself that prevents folding of the NC during translation.

The reviewer is correct that in many cases the NC does not fold until after reaching the end of the mRNA and thus stalling. This situation is to be expected given that the probability of folding after reaching the end, assuming 55 sec to reach the end and that the delay is exponential with a 71 seconds delay would be $e^{-55/71} = 46\%$. As shown in the new panel 5e, if we record the site of folding, we see that 42% of ribosomes do not fold until after codon 213, meaning they are within 10 codons of the end.

The reviewer's second concern, about the folding delay being an artifact of the assay, is addressed in the next point.

2. One obvious difference between conditions 2 and 3 is that under the latter the NC in the ribosome-nascent chain complex is under constant tension, being pulled on by a force between

3.5-4.5 pN at all times. This is a non-physiological situation, and I wonder whether this in itself could prevent folding of the NC during translation (perhaps in conjunction with the force that ratcheting of the ribosome against the mRNA might produce on the NC)? Can an experiment be designed where translation is initiated at $t=0$, but pulling force is applied only later at a time that correspond to, e.g., the emergence of the first 167 residues from the ribosome? Would any (mis)folded molecules be apparent under such conditions?

The reviewer raises a valid point and it is apparent that we did not make it clear in the text that in both conditions 2 and 3, there is always a force between 2.5-4.5 pN applied to the nascent chain. As shown in Figure 3c, RNC177 folds between 2.2-4 pN (purple circles). We chose to conduct condition 3 (translation) at such low forces, rather than higher forces where the signal would be larger and the noise reduced, in order for it to be comparable to condition 2 (stalled). It is correct that the ribosome is actively translating in condition 3 and these dynamics might be related to the change in NC behavior (the reviewer mentions ratcheting along the mRNA). These are intriguing possibilities for the underlying cause of the non-equilibrium effect we observe during active translation, although it is not clear how force from ribosome intersubunit ratcheting along the mRNA could be propagated through the NC (perhaps through contacts in the exit tunnel?). We have clarified in the text that the stalled complexes are always under tension during measurement and the translation was conducted in the same range of applied tension. This now reads: ***“We conducted translation at 4.5-3.5 pN, so that it is in the similar range of forces as probed using stalled RNCs.”***

Since we do not know when the ribosome has reached 167 without applying ~ 3 pN of force, the design the reviewer mentions would not be feasible. First, the time to reach 167 will be quite variable— although translation takes 1 amino acid per second, there is also time required for mixing and restart, as well as natural spread in the data, so we cannot predict when to raise the force to “probe” the current state. Second, we need the end point of translation to back-calculate the codon position and so we cannot, at the time of the experiment, tell the sequence position until after the data is converted. If instead we raise the force to “probe” the current state at a fixed time interval, we will initiate this probe at different positions each time and thus we will not be probing the state of interest. If we wait a long time such that we can be sure the ribosome is at or past the codon of interest, this would be equivalent to the stalled RNCs.

4. The authors propose no mechanistic explanation for the postulated folding-incompetent state of the growing NC – this is an obvious weakness of the paper.

We briefly speculated on this in the original paper but have expanded this section of the discussion. This could be due to the act of active translation itself or the orientation of the chain or tunnel. This section now reads:

“This behavior indicates that prior to the onset of folding, the newly synthesized polypeptide resides in a “trapped” unfolded state that is not folding-competent. This state may involve a set of local interactions with the ribosome surface or exit tunnel that were established during its growth in real time or, alternatively may form only in the context of actively elongating ribosome surface dynamics. For example, in real time the ribosome will have different cofactors in solution binding and interacting with its surface, as well as rotation and movement of ribosomal components. It is also possible that the chain may form local structures when it is extruded from the tunnel that are different from a random coil unfolded state. For example, small helical sections are known to fold in the tunnel and they may

prevent the folding process.”

Minor comments:

4. Please explain the difference between measurements in “active” and “passive” modes, as a help to readers not expert in single-molecule OT.

The reviewer raises an important clarification point. In our manuscript, there is no one “active mode” per se. An active mode would be any type of data collection where the traps change position as a function of time, due to user manipulation or electronic feedback. In this study, we use 1) force-extension measurements, where the tethered molecule is subjected to increasing force a user-defined pulling rate, and 2) passive mode where the traps positions remain fixed. In this latter case, a folding transition of the protein would move the tethered bead farther away from the center of the trap and this would give rise to an increase in the force acting on the tether. We have modified two sentences on page 3 to clarify this point. First, we state ***“We tethered the isolated full-length (FL) protein by attaching the N and C termini to DNA handles (Fig. 1a, b) and collected data via force-extension measurements, where the traps move apart at a fixed rate. In this measurement, applying load (increasing force) leads to unfolding of calerythrin (detected as increasing extension).”***

Later, we explain passive mode by writing: ***“We collected data in passive mode (where the traps remain stationary, as contrasted with force-extension measurements where the traps move apart at a fixed rate). In this mode, the force suddenly increases if folding occurs, and these transitions can be investigated in more detail.”***

5. In the Discussion, it is mentioned that no folding delay was seen in the real-time FRET analysis of cotranslational folding of the HemK protein (ref 13), and this is ascribed to poor synchronization in the FRET assay. The FRET assay is synchronized, however (ref 18), and a massive delay in folding such as that reported here ought to be clearly visible in the data. Of course, the HemK protein is not the same as the Ca-binding protein analyzed here, but I don't think that the FRET results can be discounted on the basis of poor synchronization. In the HemK case the NC is not being pulled on during translation, and hence not under tension.

The HemK protein assay described uses FRET and PET pairs in the nascent chain to monitor folding and SDS-PAGE to monitor translation. For the translation time course, the band in question marks the end point of translation and the percent of ribosomes that have reached this point. It is not synchronized in that the sequence positions of the ribosomes are not known. For example, the earliest time point in SDS-PAGE is around 30 seconds, when 25% of ribosomes have produced full-length product. The overall FRET signal will then derive from the 25% that are at the end and the 75% that are scattered heterogeneously along the length of the mRNA (Ref 13). Our point is that this lack of synchronization could easily make it difficult to detect deviations from equilibrium behavior. Nonetheless, it was not our intention to discount the FRET work altogether. We agree with the reviewer that a long delay of 63 seconds would likely be observed in those experiments, especially considering the multiple methods used to probe the chain's state. However, a shorter delay of even 10 seconds would be difficult to detect. Indeed, when the kinetic modeling on the data set is done in more detail (Ref 18), they find there are 2 significant ribosomal pause sites that are rate-limiting for folding. These translational pauses could obscure non-equilibrium folding effects. It could also be due to the specific substrate used. We have modified our reference to clarify this interpretation. We think that the delay observed is

unlikely to be due to the application of force, since force does not inhibit folding after the delay ends, and folding/unfolding events are reversible at this point.

5. *In Methods, it is stated that the real-time elongation experiments were done in an amino acid mix with “all 20 amino acids except arginine, cysteine, proline, tyrosine, or valine”. Does this mean that none of the amino acids were included, or that different experiments excluding different amino acids were done? Only valine exclusion is mentioned in the Ms as a way to synchronize the reaction; what were the results when omitting the other ones?*

We thank the reviewer for raising this clarification point. Stalling was performed by omitting R,C,P,Y and V. We recognize that this was phrased in a confusing manner in the manuscript and have updated this section. Although the stalling site is a triple valine repeat, none of the amino acids R, C, P, or Y were required prior to this site and so they were also omitted. Read-through is due to misincorporation and so using minimal amino acids will reduce competition with mismatched tRNAs, as well as simplifies the experiment.

6. *Figures 2-5: Please indicate the different folding/refolding events mentioned in the legends by arrows.*

7. *Figure 3a: Please explain in the legend what is different between the red and black traces.*

8. *Figure 4d: I understand that folding into the misfolded state and into the folded C domain yield peaks of different magnitudes. It would be good to have a simple shorthand explanation for how reliably one can distinguish between the two kinds of events (and distinguish them from other events such as the somewhat smaller peaks seen upstream of the “misfolded” peak in the trace). If there are more peaks that you have assigned as “misfolded” or “C domain” in this trace, please indicate them also, just to help the uninformed reader get a better feel for the assay.*

We have addressed these concerns by adding insets that show the relevant regions at higher frequency, to exemplify what a transition looks like. Within these insets we have added dashed lines that mark the transitions for each size, M vs C, to help the reader distinguish these events. We have also added arrows for some events, although we have not marked every event.

9. *Supplementary Fig. 4. In what way do the linkers differ? Please include the full sequence of the normal constructs as well as the ones with altered linker (I assume that it’s the linker in the protein that differs, not the DNA handles?). Also, please detail the conditions \pm TM in the legend.*

The linker refers to the sequence added for stalling at the valine repeat. We have clarified this in the legend (updated numbering is now Supplementary Fig. 8) and also added the exact amino acid sequence of the linker into the methods section. We have also clarified that there is a shorter SG linker present in all conditions, and in this case it is extended. We have specified in the legend that \pm TM corresponds to experiments with and without translation mix and added a reference to Supplementary Table 3, which lists the components in the translation mix.

Reviewer #3 (Remarks to the Author):

Alexander et al. use a single-molecule translation assay to study how active translation by the ribosome may affect the folding of nascent polypeptides. By comparing the folding of a given protein after translation (without ribosomes present), during arrested translation, and during active translation, they find that there are noticeable differences (most notably, delayed folding

during active translation), which suggest that the folding is not in equilibrium during translation. Furthermore, this work shows that the co-translational folding kinetics may differ from what would be expected based on studies of folding off the ribosome as well as on stalled ribosomes.

This manuscript addresses an important question, one that has proven difficult to study using traditional methods, and it contains some very interesting and beautiful results. It should be of interest to the broad readership of *Nature Communications*, and will be suitable for publication after addressing the concerns noted below.

1. Page 3, first paragraph: when discussing previous work on EF-hand family proteins, it would be a good idea to include a reference to Heidarsson *et al* PNAS 2014, which discusses Ca-dependent misfolding (of direct relevance to this work).

The work of Heidarsson *et al* finds misfolding of a similar EF-hand protein can occur from a partially folded C domain state, whereas our misfolding is more similar to that reported in calmodulin in that it occurs from the unfolded state. We agree that this is an interesting case and enriches the background and discussion even if the mechanism of misfolding differs and accordingly, we have added a reference to Heidarsson *et al* 2014 in the manuscript. It is also relevant in terms of methodology for detecting and characterizing the misfolded states, and we have emphasized also this point. The results section now reads:

“Misfolding of EF-hand proteins has been reported previously, either through mispairing EF hands or via off-pathway compaction of partially structured intermediates^{23,26}. Since for calerythrin the misfolded state forms directly from the unfolded state and involves the entire available sequence of the truncated protein, state M could involve mispaired EF hands, similar to a previously reported misfolded state of calmodulin²⁹”

2. Page 3: when discussing the pulling-curve results, it would be a good idea to mention that the length change matched what was expected for the native structure.

We thank the reviewer for this suggestion. For the full-length protein, the observed unfolding transition from pulling curves was 65 ± 2 nm, or 177 ± 5 amino acids. We have added this to the text as suggested, where it now reads: **“The unfolding transition is 65 ± 2 nm, or 177 ± 5 amino acids, as expected for the full unfolding of calerythrin, which is 177 amino acids in length.”**

We point out that for the individual domains, we did not directly measure transitions from the pulling curves due to the fast and reversible transitions. Instead we use the passive data, which has better resolution.

3. Page 3: the glutamine mutation in EF1 doesn't affect formation of FLint, but it does seem to change the unfolding from two-state by introducing an intermediate. This difference is worth mentioning.

The reviewer is correct that the D29Q mutation affects the unfolding of the FL state. Here we see the mutation has abrogated the domain unfolding cooperativity and the D29Q mutant unfolds in two domain-wise steps. Although we mentioned this in the supplementary caption, we have also added a sentence to the main text to further explain the mutation's effect. Our primary focus was on the refolding behavior, but the unfolding behavior is also affected and we should have made this clearer. This section now reads:

“The glutamine mutation in EF1 (belonging to the N-terminus domain) led to a significant change in the unfolding transition of the FL protein, which is no longer cooperative and the domains unfold in two sequential steps. However, this mutation does not affect the refolding or unfolding kinetics of the first observed intermediate, which is indistinguishable from the wild-type FL_{int} (Supplementary Fig. 1).”

4. Page 4, top: the basic results that the ribosome destabilizes structures near the N-terminal domain in the nascent chain, and that overall folding is slower with the ribosome present, are not really new, as Kaiser *et al* saw exactly the same thing previously in their work comparing folding off and on the ribosome. But the extent of the changes seems to be different here, since the destabilisation is enough to prevent folding entirely. These connections to previous work (and their significance) should be discussed explicitly.

We agree that this phenomenon has been previously reported, and although we cited this paper in the introduction, we should also have mentioned it here. In the Kaiser *et al.* paper on T4 lysozyme, folding was slowed by about a factor of 200 at a 41-amino acid linker length compared to folding in solution. By other methods, the ribosome has been observed to destabilize folded structures by up to 2 kcal/mol (Samelson *et al* 2016). In our case, the N domain stability is only -3.7 kcal/mol. If the folding rate is changed by 100-fold, the protein stability, assuming that the unfolding rate is not affected, would only be -1 kcal/mol, similar to thermal energy (0.6 kcal/mol). We have clarified this point in the results section to better contextualize this finding. It now reads:

“The isolated N domain off the ribosome folds at a rate of $k_0 = 660 \pm 370 \text{ s}^{-1}$, so we infer that the ribosome destabilizes the N-domain by reducing its folding rate or increasing its unfolding rate relative to the isolated N-domain such that it is not sampled. A decrease in folding rate of up to two orders of magnitude has been reported for other proteins such as T4 lysozyme on the ribosome, although in that case folding was reduced but not eliminated⁸. The N domain intermediate observed here has lower stability than many full proteins or domains, and the destabilization may be sufficient to prevent folding entirely.”

5. Page 4, paragraph 2: the call-out should be to Table S2, not Table S1.

We wanted to reference Table S1 to compare the size of the N and C domains (Supplementary Table 1) to the size of the misfolded state, which is reported in line but is also included in Supplementary Table 2. To clarify the comparison we are making, we have added a panel to figure 2 (Fig. 2e) which shows the passive behavior of RNC167 and RNC177, which can then be compared to figure 1d, as well as to the information in Table S1. We have updated these call-outs.

6. Page 4, paragraph 3 and following: Although most of the identifications of particular states with proposed structures in this manuscript seem well-founded, the interpretation of some of the states in the stalled complexes as misfolded is not convincing. (To be clear: the interpretation is not unreasonable, and not necessarily wrong, it's just not proven properly.) A misfolded state is typically defined as a stable or metastable state that is off the native pathway--i.e. there is a large enough barrier for transitions between that state and the native state that direct transitions from the misfolded state to the native state do not happen (otherwise, the putative 'misfolded' state would actually just be part of an alternative pathway to the native state). The problem here is that the authors have used the length change to deduce that a state is not native, but length

changes alone may not be sufficient to establish that a state is misfolded. Other explanations are possible and must be ruled out, for example it could be that the state is actually an intermediate on the native pathway that is not observed under normal circumstances but becomes detectable because of the truncation.

The authors claim that the ‘misfolded’ state is indeed off-pathway because it never transitions to the native NTD fold, but they provide no analysis proving this claim. Indeed, even without access to the full dataset, close inspection of the data presented in Fig. 3 suggests that the claim that there are no transitions from the ‘misfolded’ state to the native NTD fold is incorrect: one can see what looks like a very brief pause in the folded NTD state as the ‘misfolded’ state transitions to unfolded in a brief event at ~0.8 or 0.9 s in the record. One can also see very brief “spikes” down from the ‘misfolded’ state to a level consistent with the folded NTD at ~1 s and ~1.4 s. The short lifetime of the folded NTD state may be confounding the analysis by making it more difficult to see such transient intermediates in the folding/unfolding of the putatively ‘misfolded’ state (certainly, given the relatively low sampling rate of the data, one would not necessarily expect to see such a transient intermediate on every transition into/out of the ‘misfolded’ state).

Given the important of this putative misfolded state to the authors’ interpretation, more careful analysis of this state is needed. The authors should examine their data more carefully for signs that there are very brief intermediates during the formation of the ‘misfolded’ state (on transitions to/from unfolded) or during fluctuations out of the ‘misfolded’ state. The methods described in Yu et al PNAS 2012 for detecting transient misfolded states could be useful for this analysis. More centrally, the authors need to include some analysis of whether this putative ‘misfolded’ state can or cannot transition directly into the native fold, since that is the true indication that it is misfolded. Even if the ‘misfolded’ state forms after the NTD folds natively, it could still end up being an off-pathway misfolded state (as in the example in calmodulin observed by Stigler et al).

We appreciate the thoroughness of the reviewer’s analysis in raising this point, and agree that it is crucial to our claims. As such, we are happy to provide further evidence that this state is indeed off-pathway. Our initial manuscript based the claim that it was off-pathway on a few different pieces of evidence:

- 1) the misfolded state is not observed in our hopping trajectories of the FL protein.
- 2) the misfolded state and the N domain are mutually exclusive states for the EF123 protein, and each must unfold prior to the other state forming.
- 3) the size of the transition requires cross-domain contacts, which do not match the domain-wise folding of the FL protein.

We further speculated that it may be due to EF-hand mispairing, as has been reported for other EF hand proteins, due to the measured size and the fact that EF2 is non-canonical, while EF1 and EF3 are canonical and thus may have binding cooperativity for calcium.

The method of Yu *et al* establishes that certain intermediates are misfolded if they are only accessible from the U state, not on pathway to/from the folded state. As shown by Stigler *et al.*, a misfolded state can be off-pathway and still be accessible from an intermediate, just not leading to the FL structure. We have used the method of Yu *et al* to show that transitions to N from M are not detected and that transitions to M from the FL are not detected. This result is now added to the new Supplementary Figure 4.

With regard to the reviewer's point that transitions from M can perhaps be seen in our data, these did not give significant skew to the point spread function. Furthermore, if there was an additional state within state M ($M \rightarrow N \rightarrow M$) this would manifest in the lifetime distributions of the states involved, which would then appear non-exponential. Example cumulative distribution functions (CDFs) are shown in an added Supplementary Figure 1.

7. Page 5, top: The extrapolation of the 'misfolded' state occupancy at zero force seems suspect. Static measurements don't really provide a good basis for projecting the relative (equilibrium) occupancies during elongation because the relative stability of the different states (and the rates for leaving this partially folded state) may very well depend on the length of the polypeptide chain and hence vary with time. Given that the authors describe active ribosome measurements in the next section, this crude and somewhat misleading estimate does not seem to be needed.

We agree with the reviewer that the rates resulting from our analysis are very likely to be sequence-dependent and that this analysis is an approximation. However, since we observe the same rates at RNC167 and RNC177, we have assumed they are constant in this region. It is likely that as we translate into the linker they change, but as shown in the added panel Figure 5e, using the equilibrium rates we would expect nearly 100% misfolding prior to entering the linker region. We provide the extrapolation only to help predict what would be expected in real-time. As a compromise, we now specify the assumptions of this extrapolation in the text, and indicate explicitly that it is merely a guideline for the expected behavior during translation. On page 5 when we introduce the estimate, it now reads: ***"In real time elongation conditions, assuming the rates we have measured do not change substantially with chain length (as is the case between codons 167 and 177), we would expect on average one misfolding event per 1.5 seconds at 4.0 pN.***

8. Page 5, paragraph 3: In figure 4 c and d, it is unclear how states are being detected and assigned. The experimental errors/uncertainties look much larger than in the static ribosome measurements, for understandable reasons. The claim that the N state (NTD folded) doesn't occur is not convincing from these data. For example, in Fig. 4d there are "spikes" upward like at ~ 274 s that look like the right size for folding of N (smaller than C, as expected from the previous data). Also, are the authors assuming that even though the kinetics have apparently changed a lot from the stalled ribosome data, there are no additional (different) states formed here? More robust analysis of the data to back up the claims (or appropriate modifications of the claims to match the evidence presented) is needed.

We have added an additional section to the methods and an additional supplemental figure (Supplementary Fig. 6) to address these concerns. We also adjusted our sensitivity to ~8 ms (from 20 ms) to avoid missing short transitions. The change in our delay from 71 seconds to 63 seconds can be attributed to this change. From the stalled complex data, 8 ms sensitivity should capture ~85% of misfolding events at 4.0 pN. Although we may miss some events, this would not account for the long delay as there should be dozens of events within a 63 second time frame.

9. Page 5, bottom: if state N is present, presumably the analysis of the pathways would have to be changed.

Yes, the reviewer is correct that this would be the case.

10. Page 5, bottom: regarding the delay times τ , the authors claim that an exponential distribution (leading to a curved CDF) fits the data well, but in fact the CDF is very clearly linear--the fit is systematically off. The authors claim a very good fit quality (R^2), but the problem is that the deviations are systematic, not random, which indicates that the model is not good (it's very likely that a test such as Wald-Wolfowitz should show that the residuals to the fit are not actually random). It therefore seems that a non-exponential distribution of times is present and a different model should be used to fit the data (perhaps something like a stretched exponential?). This non-exponential distribution is actually quite interesting, as it has important implications for the origin of the delay. Certainly, the interpretation presented here that the equilibration happens via a single stochastic step (giving rise to exponential time distribution) is not well supported by the data, and a more complex model is needed. Is there some strong reason to expect that the delay *should* be exponential? If so, what are the implications of the fact that it clearly is not? This issue should be explored further in a revised manuscript.

The CDF cannot strictly speaking be a linear function as it needs to asymptotically approach 1. We expect an exponential distribution as the simplest first possibility for a kinetic process (only 1 free parameter). Due to the fact that the empirical CDF will always end at exactly 1 rather than asymptotically approach it, it is likely to exhibit skewed error at later points rather than random errors. However, it is true that other distributions are possible. And it is possible that the process leading to the delay is indeed more complex than we have assumed here and that accordingly the distributions are more complex. In the supplementary material we have provided tests for an exponential distribution (one dominant kinetic step) vs. a double exponential (parallel steps, i.e. a subset of ribosomes are slowed) and an Erlang distribution (two steps in series of similar rate). These fits are compared via an f-test and are shown in Supplementary Fig. 7.

11. Page 5 bottom/6 top: The data in Fig 5 clearly show brief "spikes" of transient folding before the delay time identified on the figure is finished. For example, in Fig 5a there is a spike at ~225 s, in 5b there is one at ~210 s, in 5c there is one at ~110 s. They look like possibly folding of N or C, but it's unclear; they just don't lead to anything stable. The events seem to be ignored entirely by the authors? They need to be discussed and incorporated into the interpretation!

In the case of real-time co-translational trajectories, the data is shown heavily downsampled in order to display it on a reasonable timescale. Real-time data is shown at 27 Hz, whereas the equilibrium folding is shown at 250 Hz; generally only 3 seconds of stalled RNC data are shown at a time, compared to tens to hundreds of seconds for real-time. Nonetheless, for all the data presented here, the full recording frequency was used to identify transitions and calculate their sizes. In Figure 5 we wished to display the observed delay time, thus requiring us to show a long-time period. In light of the reviewer's comment, we have now included additional insets to each panel where the data are zoomed to the folding regions and shown at higher frequency (267 Hz) to clearly display the distinction between signal and noise. We have updated the caption to reflect these changes and emphasize the new display frequency.

The spikes the reviewer mentions do not have significant residence time in the higher-force state when viewed at full bandwidth and are attributable to noise, with the exception of one spike in panel 5b. This zoomed in region shows that there is likely a brief transition, but it is smaller than both the M state or the N state and cannot be assigned. This type of event is rare and does not reoccur, but we have nevertheless clarified it in the text.

12. Page 7, top: *If the “unfolded” state has a much longer lifetime in the presence of the active ribosome, then clearly it can’t truly be the same “unfolded” state as in the absence of the ribosome--there must be some interaction or process that differentiates it with the ribosome present and slows down the folding.*

We agree that there is some change present to the unfolded state. We have added some speculation as to what this may be prior to the discussion. This part of the manuscript now reads:

“This behavior indicates that prior to the onset of folding, the newly synthesized polypeptide resides in a “trapped” unfolded state that is not folding-competent. This state may involve a set of local interactions with the ribosome surface or exit tunnel that were established during its growth in real time or, alternatively may form only in the context of actively elongating ribosome surface dynamics. For example, in real time there will be local dynamics of ribosomal components as factors bind/unbind and ribosomal elements alter their conformation. It is also possible that the chain may form local structures when it emerges from the tunnel that are different from a random coil unfolded state, that may be folding-incompetent. For example, small helical sections are known to fold in the tunnel that may prevent the folding process.”

13. Page 7: *These experiments provide a very nice way to probe how the dynamic ribosome influences the folding, but of course they don’t recapitulate the situation in the cell (e.g. lack of crowding, which could have an important effect on the relative stability of partially folded and/or misfolded states). Some brief acknowledgement of this issue would be helpful in the Discussion section.*

We agree that other cellular factors could influence nascent chain stability. When we introduce the translation mix, we now mention that we do not have the same degree of crowding as present in the cell. We also have not added chaperones. This section of the manuscript now reads:

“we find an average elongation rate of 1.0 ± 0.3 amino acids/s, consistent with other in vitro translation experiments using the commercial PURExpress system^{32,33}. We point out that our in vitro system does not perfectly recreate the cellular environment— for example, it does not include chaperones and does not mimic the extent of molecular crowding in the cell.”

14. Page 10, end of first paragraph: *the silver-staining data should be added to SI.*

We have added the silver staining data to Supplementary Fig. 5.

15. Page 10, paragraph 4: *please clarify which parameters were treated as fixed and which were used as fitting variables. The wording seems to suggest that for the DNA handles, the persistence length and stretch modulus were fixed but the contour length was fitted, which seems backwards in some sense (the contour length is the one parameter that is known precisely for the handles).*

We thank the reviewer for raising this clarification point. We have now clarified our fittings in the methods section. The persistence length and stretch modulus were fit using DNA-only conditions and a fixed contour length. Once determined in our buffer conditions, these parameters were treated as fixed for the conversion of passive data to contour length and amino acids. This part of the manuscript now reads:

“To convert force data into amino acids, we used a combination of two worm-like chain models. For the DNA, we used an extensible WLC model with persistence length $P = 35$

nm and stretch modulus $S = 1200$ pN. These WLC parameters were determined by pulling DNA-only tethers in the same buffer conditions to fit the persistence length and stretch modulus, while using a known fixed contour length. The resulting parameters are as should be expected for buffer conditions at high magnesium and polyamine⁴¹ concentrations. For the polypeptide contribution, we used a worm-like chain model with a persistence length of 0.6 nm.”

16. *For all the figures containing pulling curves, please show the worm-like chain (WLC) fits to each different state, to help assess the quality of the data. Also, please add some of the data at low force so that the ‘knee’ of the force-extension curve (which is most reflective of persistence length) can be seen.*

We have added overlays of the WLC to the pulling curves. Since we were collected folding data and all folding completed above 2 pN, we do not have data for the very low force regions of the pulling curve that the reviewer mentions. There are some deviations at low forces for the WLC on the surface of the ribosome, which has previously been reported by Kaiser *et al* (2011).

17. *In the Methods section, please describe how it was ensured that only single molecules were being measured. On page 11 it is implied that the contour length alone is used, but that is known to be an imperfect indicator of single vs multiple tethers between the beads.*

We used both contour length and curvature to ensure single molecules were analyzed. For most of the constructs there is also a clear signature of unfolding. We have now stated this explicitly in the Methods section.

18. *Page 12, top: please clarify what property is calibrated after each experiment.*

The trap stiffness and bead offset files due to trap crosstalk were calibrated for each pair; this is now stated explicitly in the Methods section, which reads now: ***The trap stiffness and bead offset files due to trap crosstalk were calibrated for each bead pair after each experiment.***

19. *Page 12, end of Methods: there is growing acknowledgement that the criterion $p < 0.05$ is too weak to be reliable, as it yields many false positives (see e.g. Calquhoun 2014). The authors would be advised to treat results close to $p \sim 0.05$ (as for one of the results in Table S1) with skepticism as to whether it is truly significant.*

We agree with the reviewer that the p-value is an imperfect metric. In Table S1, we are comparing the domain truncations to the full-length intermediate. Fortunately, we have multiple variables for comparison (folding rate, unfolding rate, size, distance to transition state for both folding and unfolding processes) and thus, we use all of them in combination to help assign the intermediate, rather than relying on one value.

20. *Figure 2C: Please clarify if these curves are unfolding or refolding. Note that there are clearly some dynamics going on in construct 167, but they are not mentioned/discussed clearly in the manuscript. WLC fits should be added to ascertain if the protein is purely unfolded as claimed, or if there are rapid or non-cooperative transitions present (which can be quite subtle, producing curves that ‘slide’ from one WLC to another as seen in Zoldak PNAS 2013 or Solanki PRL 2014).*

All of the curves are shown for refolding and we have added this clarification to the caption of Figure 2. We have added the WLC fits as well. We apologize for not making RNC167 behavior clear in the manuscript, it has the same kinetics as RNC177. We used RNC177 kinetics in the figures for consistency. We have now expanded this point in the text. It now reads:

“Significantly, the ribosome decelerates its folding kinetics at codons 167 and 177 by 10^4 -fold and accelerates its unfolding rate by 90-fold at zero force compared to the free truncation EF123 (Fig. 3b, c, Supplementary Table 2).”

21. *Figure 2D: the color- and dash-coding of the curves are difficult to understand. Perhaps a rainbow color-coding could be used?*

We have added additional labels above the figure to show the overlaid WLCs, and changed the line formatting for clarity.

22. *Fig 3A: Please add dotted lines showing the position expected for the intermediate state proposed here. Fig 3B: Please clarify whether these pulls are unfolding or refolding. Also, the unfolded states appear to be slightly misaligned. Fig 3C: please explain the figure symbols in the caption.*

We have updated the figure to address all three of these concerns. We have added arrows that mark transitions. We have also added a new supplementary figure, Supplementary Fig. 4, which addresses the intermediate in more detail.

23. *Fig S1: The caption in (B) says the fully folded state is not stable, but this seems incorrect since it forms when the force is ramped down. Please clarify or correct.*

For the WT full-length protein, the folded structure is long-lived. We do not observe transitions out of this state in passive unless we change the trap position. For the mutant, at the same trap positions we can see transitions out of the folded structure (now numbered Supplementary Figure 2b), and therefore, it is *less* stable under 6.8 pN, relative to the WT. We updated the caption to now read instead ***“In addition, the fully folded state shows new partial unfolding transitions in passive mode, which were not observed for the WT, (the downwards jumps from the state at 6.8 pN).”***

24. *Fig. S1: For all the mutants, please list the measured length changes and how they compare to the expected length changes (either in the caption or in a table), and of course show the WLC fits.*

We have updated the figure (now numbered Supplementary Fig. 2) with the WLC for a protein of 177 amino acids and our handles, as well as the handle-only WLC. Due to the many small transitions that are present in all of the mutants, which result in many small rips being detected, we did not measure an overall transition size from the pulling curves.

25. *Fig S2: Since the force is varying throughout these measurements, please clarify what is meant by “at 3.5 pN”.*

To compare the behavior of these RNCs, we selected data in which the unfolded state occurs near 3.5 pN; the folded states would be seen at higher force. We have updated the caption accordingly.

26. In the SI, please add a figure showing the distributions of lifetimes for each state, and any exponential fits to them, so that the readers can assess the data used to evaluate the kinetic rates.

We agree that the CDFs are valuable to assess the quality of the data. For each construct, we have hundreds of CDFs that are all fit individually for each recording at each trap position, then binned to produce the $\ln(k)$ vs force plot. For example, for the FL protein we have 116 CDFs for both folding and unfolding. As this may be an unreasonable number to include in the SI, we have instead elected to show example CDFs for each construct from near the mid-force. These are now shown in Supplementary Fig. 1.

27. Tables S1, S2: rather than showing the “size” of the states in amino acids, which is an inferred quantity, it would be better to report the measured quantity, length change, and then compare to the length change expected from the proposed structures. Please describe in the Methods how the conversion between length changes and amino acids is done.

Since not many readers may be familiar with changes in nm, we think it is more appropriate to show the converted value in amino acids, for clarity. We used a contour length-to-amino acid conversion value of 0.365 nm/aa, which we have now added in the methods section.

28. Tables S1, S2: the errors in the rates do not seem to be treated properly. Given that they are obtained from a linear fit to $\ln(k)$, the rates should be reported in exponential notation with the errors in the exponent. This approach should get rid of the “unphysical” errors the authors mention that they obtained in some cases.

Previously we used the errors from the exponent as upper and lower bounds, but we have now reported the errors directly in the exponent as the reviewer suggests. We still show the central fit value for ease of comparison, then the error range on the following line.

29. The authors may wish to expand their discussion by outlining why calerythrin differs from calmodulin folding (in terms of intermediates) as found previously by Stigler et al.

Our folding is more similar to NCS1 in that it is domain-wise passing through the C domain. Calerythrin and NSC1 only have 3 binding sites, rather than 4, with the higher affinity sites in the C domain. We have added this to the discussion of the results for the FL protein and also when discussing the misfolded state, as both of these observations differ from calmodulin’s behavior.

Reviewer #4 (Remarks to the Author):

*The manuscript describes the folding of the nascent polypeptide chain in and outside the ribosome tunnel. The object for this research is well chosen, the multi-domain protein calerythrin from bacterium *S.erythraea*. It consists of four EF hands, two in the N domain and two in the C domain. The paper describes the interaction between the rate of synthesis and the folding for the whole protein and the two domains. All this is observed in single molecule experiments using optical tweezers.*

The manuscript is carefully written providing good evidence for the conclusions. The results are new showing the interaction of peptide synthesis and folding in time. In this respect the paper is showing important details of folding and misfolding of the whole protein and the N and C terminal domains. The understanding of protein synthesis and folding is of principal interest in

cell biology.

The manuscript can be published as it is. However, it should be clearly stated that this in vitro system does not contain the ribosomal trigger factor or other chaperons like Dnak or GroEL. On the other hand it would have been quite interesting to have seen how things are changing in the presence of these chaperons. The authors mentioned that they observed misfolding. What is the structure of these misfolded nascent polypeptides? Maybe they could shortly mention interesting NMR investigations of C.M. Dobson et al. who found that an initial N terminal nascent chain showed a β -strand which converted to a helical structure when the chain became longer. Maybe the authors have evidence about the type of structure of these misfolded polypeptide chains.

In response to the reviewer's comments, we have now added a statement that we do not have chaperones present in our system. We include a reference to the work of Cabrita *et al* (corresponding author CM Dobson) which shows via NMR that the C domain of a multidomain protein is mostly unstructured but when given sufficient length a native-like structure is formed in the N domain. We speculate in the text that the misfolded state may be due to EF-hand mispairing, although our evidence is indirect. This section now reads:

“This observation suggests that the misfolded state is off-pathway for the nascent chain, involving non-native interdomain contacts that preclude the correct folding of the individual domains. Misfolding of EF-hand proteins has been reported previously, either through mispairing EF hands or via off-pathway compaction of partially structured intermediates^{24,28}. Since for calerythrin the misfolded state forms directly from the unfolded state and involves the entire available sequence of the truncated protein, state M could involve mispaired EF hands, similar to a previously reported misfolded state of calmodulin³². EF2 is also a noncanonical EF hand that does not bind calcium²², potentially allowing EF3 folding to preempt EF2 folding.”

Reviewers' Comments:

Reviewer #1:

Remarks to the Author:

The authors have addressed my concerns. Their edits provide more context and nuance that will benefit readers. This is an important contribution to the field, so I recommend publication, provided the authors just address two very minor issues.

line 282: "Such long delay could also allow time for chaperones to...". Should this read "Such A long delay ..."? The authors should feel free to ignore this suggestion if they disagree.

line 303: "we would expect to be able to predict the co-translational behavior;" The word 'continuous' should be inserted to disambiguate from co-translational behavior on stalled ribosomes. For example: "we would expect to be able to predict the continuous co-translational behavior;"

Reviewer #2:

Remarks to the Author:

A missing word in legend, Fig. 5e: "where first occurred" should probably read "where folding first occurred".

Reviewer #3:

Remarks to the Author:

The authors have provided reasonable responses to almost all of the reviewers' concerns. Only two issues remain, in my view:

1) The effect whereby truly co-translation folding is delayed compared to folding off the ribosome or on stalled ribosomes is the central result of the manuscript, so it's somewhat unsatisfying that there is no explanation for the effect. One possibility that comes to mind is that calerythrin contains a proline, and proline isomerization involves timeframes on the order of tens of seconds, similar to the 'waiting time' of ~ 60 s seen here. Could some isomerization step be involved in the delayed folding? If not, it should be ruled out in the discussion; if it can't be ruled out, then it should be listed as a possibility.

2) Because the authors have not been able to establish the origin of this central effect, they should be more cautious in the claims they make about the effect, in particular whether it is a general problem or something that is specific only to some subset of proteins. What the authors have shown is that the folding on a translating ribosome *can* be out of equilibrium (as demonstrated by the particular example of calerythrin), not that it always is (which is the current implication of their claims). For example, statements in lines 296–305 imply that this effect is general for all proteins and should be hedged to acknowledge that it's unclear how general this effect is.

Rev#1

The authors have addressed my concerns. Their edits provide more context and nuance that will benefit readers. This is an important contribution to the field, so I recommend publication, provided the authors just address two very minor issues.

line 282: "Such long delay could also allow time for chaperones to...". Should this read "Such A long delay ..."? The authors should feel free to ignore this suggestion if they disagree.

line 303: "we would expect to be able to predict the co-translational behavior;" The word 'continuous' should be inserted to disambiguate from co-translational behavior on stalled ribosomes. For example: "we would expect to be able to predict the continuous co-translational behavior;"

We have made the two changes suggested by the reviewer.

Rev#2

A missing word in legend, FFig. 5e: "where first occurred" should probably read "where folding first occurred".

We have fixed the omission that the reviewer pointed out.

Rev#3

The authors have provided reasonable responses to almost all of the reviewers' concerns. Only two issues remain, in my view:

1) The effect whereby truly co-translation folding is delayed compared to folding off the ribosome or on stalled ribosomes is the central result of the manuscript, so it's somewhat unsatisfying that there is no explanation for the effect. One possibility that comes to mind is that calerythrin contains a proline, and proline isomerization involves timeframes on the order of tens of seconds, similar to the 'waiting time' of ~ 60 s seen here. Could some isomerization step be involved in the delayed folding? If not, it should be ruled out in the discussion; if it can't be ruled out, then it should be listed as a possibility.

We considered the possibility that the delay is due to a proline isomerization. However, we do not believe that this hypothesis holds in this case. First, the N domain of the protein does not contain any proline and therefore issues of proline isomerization could not be involved in the

lack of folding we observe for the N domain. There is one proline in this protein that belongs to the C domain (residue 102), however, in the folded structure this residue appears in the trans conformation, which is also the conformation in which this amino acid is incorporated at the peptidyl transfer center. We do appreciate however the reviewer comment about this possibility and we have now added this point into the discussion since it is important to consider. This section now reads:

“This behavior indicates that prior to the onset of folding, the newly synthesized polypeptide resides in a “trapped” unfolded state that is not folding-competent. Although there is a proline in the sequence of the C domain, this trapped state is unlikely to be due to isomerization as the proline in the NMR structure is in the *trans* configuration²⁰, the orientation in which amino acids are added at the peptidyl transfer center. Instead, this state may involve a set of local interactions with the ribosome surface or exit tunnel”

*2) Because the authors have not been able to establish the origin of this central effect, they should be more cautious in the claims they make about the effect, in particular whether it is a general problem or something that is specific only to some subset of proteins. What the authors have shown is that the folding on a translating ribosome *can* be out of equilibrium (as demonstrated by the particular example of calerythrin), not that it always is (which is the current implication of their claims). For example, statements in lines 296–305 imply that this effect is general for all proteins and should be hedged to acknowledge that it’s unclear how general this effect is.*

We agree that we do not know how general this effect is and have modified the sentences mentioned by the reviewer to include this point. It now reads: “Our results show that comparing equilibrium-derived folding rates from stalled RNCs with translation rates is not always sufficient to predict co-translational folding of a polypeptide. Indeed, in this case we find that the peptide product must be observed in real-time as it is being generated, since other processes may also be at play that are obscured in stalled complex measurements.”

Sincerely,

Carlos Bustamante